# The parent-of-origin lncRNA *MISSEN* regulates rice endosperm development

Yan-Fei Zhou[1,3], Yu-Chan Zhang [1,3], Yu-Meng Sun[1,3], Yang Yu [1], Meng-Qi Lei[1], Yu-Wei Yang[1], Jian-Ping Lian[1], Yan-Zhao Feng[1], Zhi Zhang[1], Lu Yang[1], Rui-Rui He[1], Jia-Hui Huang[1], Yu Cheng[1], Yu-Wei Liu[1] & Yue-Qin Chen [1,2 ✉]

The cereal endosperm is a major factor determining seed size and shape. However, the molecular mechanisms of endosperm development are not fully understood. Long noncoding RNAs (lncRNAs) function in various biological processes. Here we show a lncRNA, *MISSEN*, that plays an essential role in early endosperm development in rice (*Oryza sativa*). *MISSEN* is a parent-of-origin lncRNA expressed in endosperm, and negatively regulates endosperm development, leading to a prominent dent and bulge in the seed. Mechanistically, *MISSEN* functions through hijacking a helicase family protein (HeFP) to regulate tubulin function during endosperm nucleus division and endosperm cellularization, resulting in abnormal cytoskeletal polymerization. Finally, we revealed that the expression of *MISSEN* is inhibited by histone H3 lysine 27 trimethylation (H3K27me3) modification after pollination. Therefore, *MISSEN* is the first lncRNA identified as a regulator in endosperm development, highlighting the potential applications in rice breeding.

[1] Guangdong Provincial Key Laboratory of Plant Resources, State Key Laboratory for Biocontrol, School of Life Science, Sun Yat-Sen University, 510275 Guangzhou, China. [2] MOE Key Laboratory of Gene Function and Regulation, Sun Yat-sen University, 510275 Guangzhou, China. [3]These authors contributed equally: Yan-Fei Zhou, Yu-Chan Zhang, Yu-Meng Sun. ✉email: lsscyq@mail.sysu.edu.cn

The endosperm of cereal grains is an important source of human nutrition, and an important tissue for studying molecular mechanisms and metabolic processes. Endosperm development begins with an initial syncytial phase followed by a cellularization phase[1,2]. During the syncytial phase, the endosperm nuclei undergo multiple rounds of mitosis without cytokinesis, producing a multinucleate cell[3]. This syncytial cell then forms individual cells during cellularization[4,5].

The transition between syncytium and cellularization is important for determining final seed size and is a model for studying the cell cycle and cytokinesis[6,7]. Molecular genetic studies on early endosperm development have mainly been performed in Arabidopsis thaliana and have led to a variety of interesting findings, including the involvement of genomic imprinting and epigenetic mechanisms[7–9], auxin signaling[10,11], microtubule[12], and the identification of transcription factors[13] that negatively control cellularization. In rice (Oryza sativa), only a few mutant-based molecular studies on early endosperm development have been conducted. Hara et al. [14] revealed that the rice SNF2 family helicase ENDOSPERMLESS 1 (ENL1) regulates syncytial endosperm development, suggesting that helicase family protein (HeFP) is essential for endosperm cellularization. OsLFR, an ortholog of the Arabidopsis SWI/SNF chromatin-remodeling complex (CRC) component LFR, regulates rice endosperm and embryo development. oslfr endosperm had fewer free nuclei, abnormal and arrested cellularization[15]. MADS box transcription factors MADS78, MADS79 regulate endosperm cellularization[16] and OsGCD1, a highly conserved homolog of Arabidopsis GAMETE CELLS DEFECTIVE1 (GCD1), affects endosperm free nucleus positioning[17]. However, our knowledge regarding to the molecular mechanisms that enable the extensive development of endosperm in cereals is still lacking, especially at the early stages.

Long noncoding RNAs (lncRNAs), the noncoding RNAs that are longer than 200 nucleotides (nt), function in virtually every biological process in mammals and plants[18,19]. In contrast to the well-established mechanism of microRNA action, which is based on seed sequence base-pairing[19], the mode of action of lncRNAs remains to be fully understood. Moreover, lncRNAs are transcribed from fast-evolving genes, which might be important for regulating species-specific processes. For example, the lncRNAs COLDAIR and COOLAIR regulate flowering under cold temperatures[20,21]. Also, the lncRNAs LONG-DAY-SPECIFIC MALE-FERTILITY-ASSOCIATED RNA (LDMAR)[22] and PHOTOPERIOD-SENSITIVE GENIC MALE STERILITY 1 (PMS1T)[23] are required for normal pollen development under long-day conditions and LEUCINE-RICH REPEAT RECEPTOR KINASE ANTISENSE INTERGENIC RNA (LAIR) regulates grain yield[24]. However, a role for lncRNAs in plant endosperm development has not been reported yet. We asked whether any lncRNAs are involved in this process, and if so, what their underlying functions and mechanisms are.

We previously performed a genome-wide screen and functional analysis of lncRNAs in rice and identified a set of lncRNAs that are involved in sexual reproduction[25]. We identified a mutant with a T-DNA insertion in an lncRNA (XLOC_057324), resulting in a low fertility phenotype. In this study, we uncovered the role of XLOC_057324 in endosperm development, revealing that the XLOC_057324 locus is maternally expressed and repressed by histone H3 lysine 27 trimethylation (H3K27me3) after pollination. We showed that knockdown of the lncRNA could facilitate the progress of nucleus division and cellularization at early-stage endosperm development and produce slightly larger seeds compared to WT. Contrast, overexpression of XLOC_057324 during postembryonic development inhibited endosperm development and led to abnormally mis-shaped seeds

with a prominent dent and bulge. We therefore named this lncRNA MISSEN. We further demonstrated that MISSEN functions by hijacking a HeFP, leading to abnormal cytoskeleton polymerization during endosperm development. The MISSEN–HeFP pathway has potential applications in breeding for improved grain yield in rice.

## Results

### The lncRNA MISSEN (XLOC_057324) affects the seed development. 

We previously identified a T-DNA insertion mutant with a low-fertility phenotype. The T-DNA insertion was at the second intron of lncRNA XLOC_057324 (Fig. 1a) and the seed setting rate was shown in (Fig. 1b). To uncover what causes the low seed-setting rate of the mutant, we systematically investigated the mutant plants. We found no apparent differences between the mutant and wild-type plants (WT) at the vegetative stage (Supplementary Fig. 1a) or at the reproductive stage, including the pollen gains, and embryo sac (Supplementary Fig. 1b). The pollen grains of the T-DNA insertion mutant plants also germinated effectively at the stigma (Supplementary Fig. 1c). These results excluded the possibility that the low seed-setting rate was caused by male or female fertility. We therefore examined seed development. Unexpectedly, when we observed the seeds of the mutant plants, we found that only 30.6% of the mature seeds were normally developed, 35.6% exhibited a dent and bulge in the seed shape (Fig. 1c and d) and 33.8% exhibited even more severe seed defect phenotype with less endosperm (Fig. 1d, Supplementary Fig. 1d). Therefore, we named the XLOC_057324 lncRNA MISSEN (MIS-SHAPEN ENDOSPERM). These findings suggested that the MISSEN lncRNA might participate in seed development. By measuring the expression level of MISSEN, we found that the T-DNA insertion was an activating mutation (Fig. 1e), with higher lncRNA levels in the mutant compared with the parental line.

MISSEN is located on chromosome 8 and this locus produces four variants (Fig. 1a). However, only variant 1 (MISSEN 1) showed a high expression level, the other three variants showed an extremely low level (Fig. 1f); thus, we mainly focused on MISSEN 1 (referred to as MISSEN hereafter). We performed rapid amplification of cDNA ends (RACE) to experimentally validate the 5′ and 3′ ends of MISSEN (Supplementary Fig. 1e). We also analyzed the protein coding ability of MISSEN using the Coding Potential Calculator (CPC, http://cpc.cbi.pku.edu.cn/programs/run_cpc.jsp)[26]. MISSEN showed a very low coding potential, similar to those lncRNAs PMS1T[23] and COLDAIR[20] which demonstrated no protein coding potential (Supplementary Fig. 2a). Temporary and spatial expression pattern analysis showed that MISSEN was highly expressed in pistil (Supplementary Fig. 2b).

To validate that the abnormal shape of the seed was indeed caused by MISSEN, we generated transgenic lines with reduced MISSEN levels (MISSEN-RNAi) or increased levels (MISSEN-OX) plants (Supplementary Fig. 2c). After obtaining the T₃ generation of the transgenic plants, we analyzed the phenotypes of the MISSEN-RNAi and MISSEN-OX transgenic plants. The MISSEN-RNAi and MISSEN-OX plants appeared similar to the WT at vegetative stages and at the early reproductive stage (Supplementary Fig. 2d); the pollen grains and embryo sac of the MISSEN-OX and MISSEN-RNAi plants developed normally and their pollen grains germinated normally (Supplementary Fig. 2e–g). MISSEN-RNAi exhibited a slightly larger embryo sac (Supplementary Fig. 2g, h), 11.4 ± 6.8% of the MISSEN-RNAi pistils had three stigmas (Supplementary Fig. 2i and j). However, when observing the panicles of the MISSEN-OX plants, a strikingly increased rate of shriveled seeds was found when compared to

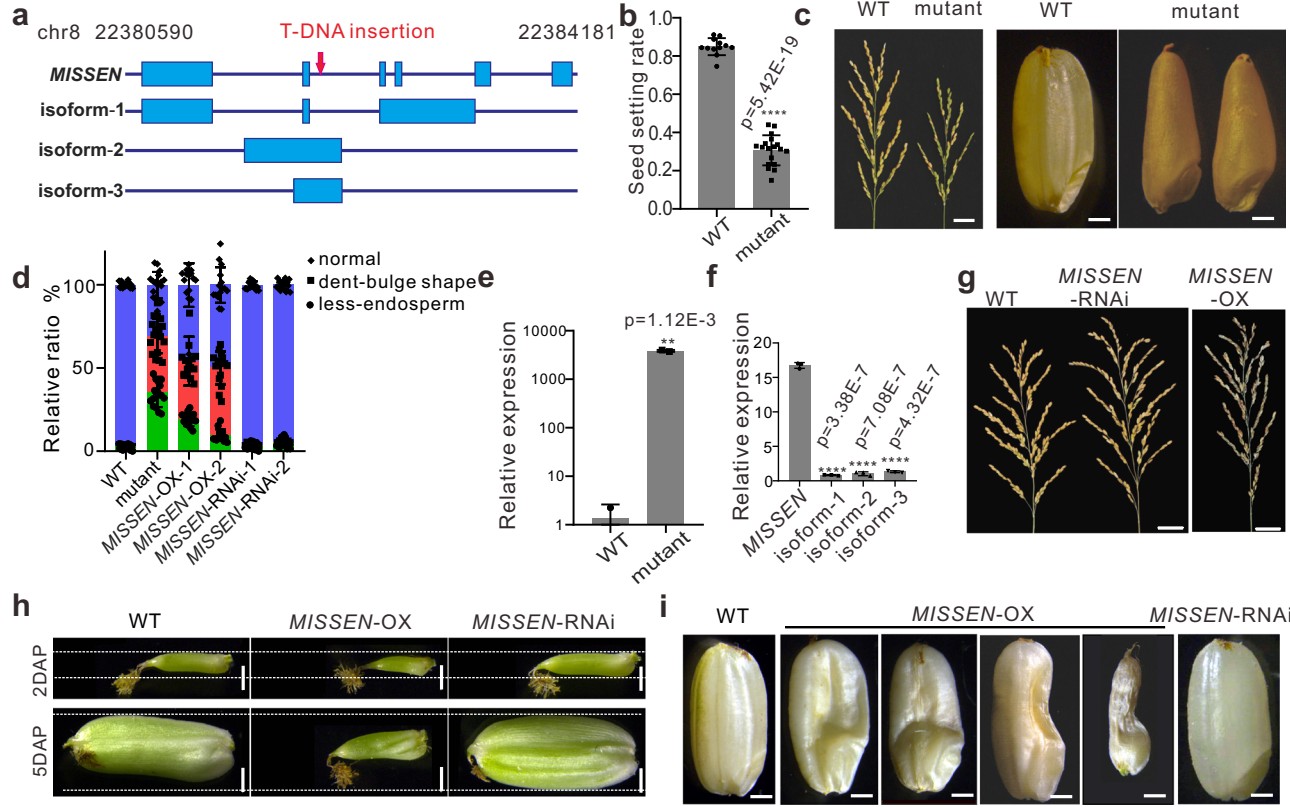

**Fig. 1 Phenotype analysis of *MISSEN* mutant plants. a** The gene structure of the lncRNA *MISSEN* and its three isoforms. The T-DNA insertion site of the mutant is indicated by the red arrow. **b** The seed setting rate of the T-DNA insertion mutant. **c** The panicles and seeds of WT and mutant plants. Scale bars, 3 cm for panicles and 1 mm for seeds. **d** The ratio of abnormal seeds in WT, mutant, *MISSEN*-OX-1, *MISSEN*-OX-2, *MISSEN*-RNAi-1, and *MISSEN*-RNAi-2 plants. Values are the means ± SD ($n = 19, 18, 15, 15, 16, 15$ plants). **e** The relative expression level of the isoforms of *MISSEN*. Values are the means ± SD ($n = 3$ replicates, normalized against *ACTIN2*); **f** The relative expression of *MISSEN* in the WT and the mutant. Values are the means ± SD ($n = 3$ replicates, normalized against *ACTIN2*); **g** The panicles of WT, *MISSEN*-RNAi, and *MISSEN*-OX plants. Scale bars, 3 cm; **h** The developing seeds of the WT and the different transgenic plants at day 2 and day 5 after pollination. Dotted white lines highlight the width of *MISSEN*-RNAi caryopsis. Scale bars, 1 mm; **i** The mature seeds of WT, *MISSEN*-RNAi, and *MISSEN*-OX plants. Scale bars, 1 mm. Significant differences were identified at the 5% (*), 1% (**), 0.1% (***) and <0.01% (****) probability levels using two-tailed paired *t*-test.

WT plants (Fig. 1g), which was similar to that of the T-DNA insertion mutant plant (Fig. 1c), indicating that the *MISSEN* lncRNA indeed regulates seed development.

We further observed grain development and the phenotypes of the different lines. The *MISSEN*-RNAi seeds grew more rapidly than the WT seeds (Fig. 1h, Supplementary Fig. 2k and l). However, similar to the T-DNA mutant, only 49.9% and 45.8% of the mature grains were normally developed in two individual lines, respectively, approximately 50.1% and 54.2% of the *MISSEN*-OX plants abnormally developed after pollination, and showed an irregular shape with defective endosperm (Fig. 1di and Supplementary Fig. 3a). Taking these findings together with the observations of the T-DNA insertion activated mutant, we concluded that *MISSEN* might function in regulating seed growth, specifically endosperm development.

**MISSEN negatively affects endosperm development**. We next investigated how the seed shape was affected in *MISSEN*-OX and *MISSEN*-RNAi plants. We systematically investigated seed development beginning from day 1 to day 7 after pollination. Morphological differences became evident at day 2 after pollination (DAP) between *MISSEN*-RNAi plants and WT plants (Fig. 1h); the *MISSEN*-RNAi seeds grew faster than those of the WT, which was particularly obvious at 3 DAP (Supplementary Fig. 3b and c). In the mature stage, the seeds of *MISSEN*-RNAi were slightly larger compared to WT (Fig. 1i and Supplementary

Fig. 3d). In contrast, the *MISSEN*-OX line showed an opposite phenotype (Fig. 1h, Supplementary Fig. 3b–d). At 7 DAP, the *MISSEN*-OX endosperm grew irregularly and the size of caryopses was smaller than that of WT and *MISSEN*-RNAi (Supplementary Fig. 3b). Most of the *MISSEN*-OX seeds carried a developed embryo (Supplementary Fig. 3e), and the embryos could germinate normally (Supplementary Fig. 3f). Therefore, we concluded that *MISSEN* regulates grain development primarily by repressing endosperm formation.

We next performed paraffin sectioning and confocal laser-scanning microscopy (CLSM) to observe developing caryopses that were stained with Eosin B. The defects in *MISSEN*-OX endosperm development started at the syncytial stage. At 1 DAP in the WT, the numbers of nuclei rapidly increased over time, all nuclei, together with their surrounding cytoplasm, localized to the periphery of the embryo sac (Fig. 2a). However, the nuclei distribution and number were abnormal in *MISSEN*-OX plants in that irregular distribution was observed (Fig. 2a, white arrows indicating the nuclei), some nuclei were suspended in the middle of the embryo sac (Fig. 2a). The nucleus number of *MISSEN*-OX plants is less than that of WT plants (Fig. 2b). At 2 DAP, less nucleus was also observed (Fig. 2b), and the irregular distribution was more obvious and a prominent dent was found in the right upper of embryo sac in *MISSEN*-OX plants (Fig. 2a). Propidium iodide (PI) staining supported this observation (Fig. 2c). At 3 DAP, cellularization progressed in the WT gradually from the

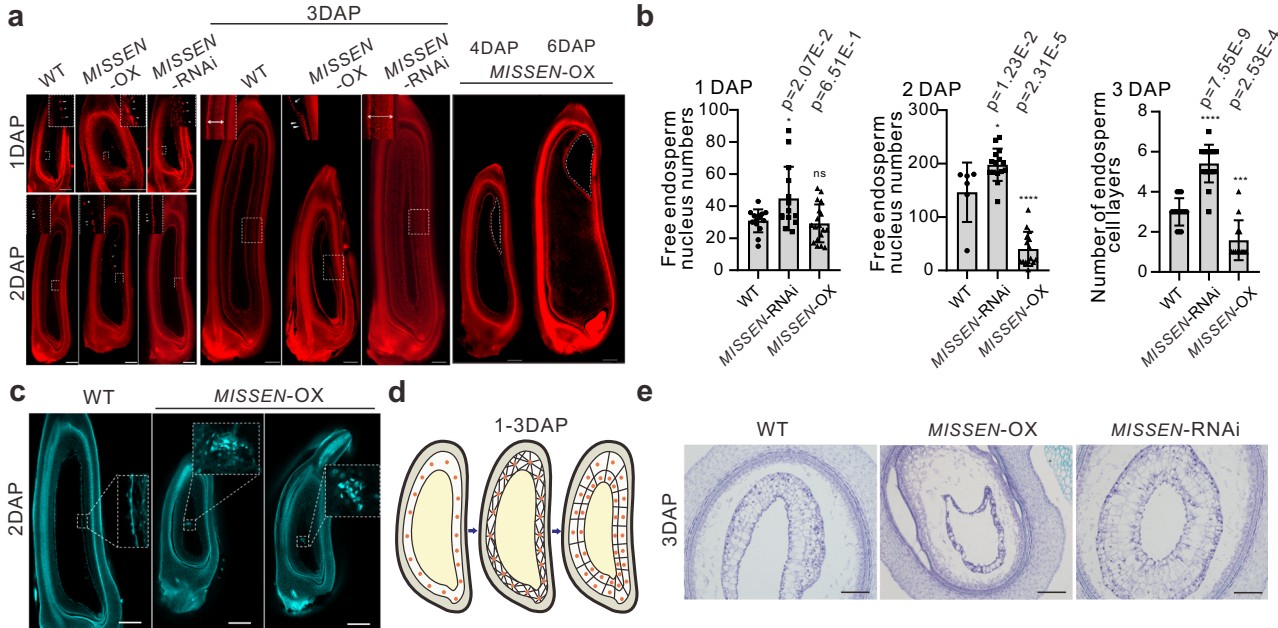

**Fig. 2 The lncRNA *MISSEN* regulates endosperm development. a** Confocal laser scanning microscopy of developing caryopses stained with Eosin B in WT, *MISSEN*-RNAi, and *MISSEN*-OX plants. The endosperm nuclei were shown in the magnified images. The cavities in *MISSEN*-OX endosperm at 4 and 6 DAP are indicated by dotted lines. Scale bars, 50 μm. The experiment was repeated three times with similar results and one representative result was shown. **b** The number of free endosperm nuclei in WT, *MISSEN*-RNAi, and *MISSEN*-OX plants at 1DAP (*n* = 14, 14, 18 caryopsis) and 2DAP (*n* = 6, 15, 15 caryopsis), and the number of endosperm cell layers in WT, *MISSEN*-RNAi, and *MISSEN*-OX plants at 3DAP (*n* = 14, 17, 12 caryopsis). Values are the means ± SD. **c** Confocal micrographs of caryopses at 2 DAP stained with Propidium Iodide in WT, *MISSEN*-RNAi, and *MISSEN*-OX plants showing the abnormal nuclei distribution in *MISSEN*-OX plants. The insets show the enlarged images. Scale bars, 200 μm. The experiment was repeated three times with similar results and one representative result was shown. **d** The schematic diagram of endosperm cellularization during 1–3 DAP; **e** Paraffin transverse sections of the caryopses at 3 DAP of WT, *MISSEN*-RNAi, and *MISSEN*-OX plants. Scale bars, 200 μm. The experiment was repeated three times with similar results and one representative result was shown. Significant differences were identified at the 5% (*), 1% (**), 0.1% (***) and <0.01% (****) probability levels using two-tailed paired *t*-test.

periphery towards the center of the endosperm, and formed 2–3 layers of cellularized endosperm cells (Fig. 2a, b and d). Figure 2d is the schematic diagram of endosperm cellularization during 1–3 DAP. In the *MISSEN*-RNAi plants, 6–7 layers of cellularized endosperm cells formed. Strikingly, the *MISSEN*-OX endosperm only has one layer of cellularized endosperm cells; walled compartments were irregularly formed (Fig. 2a, b and e). We also observed a "cavity" where endosperm did not develop further and collapsed in *MISSEN*-OX endosperm from 3 to 6 DAP (Fig. 2a). This phenomenon might explain the mis-shaped endosperm phenotype.

Taken the results together, we can conclude that the illegitimate nucleus division, distribution and cellularization occurred in the *MISSEN*-OX endosperm; *MISSEN* negatively affects endosperm development.

**MISSEN binds a HeFP**. The above results showed that *MISSEN* participates in endosperm development. We next sought to determine the underlying molecular mechanism by which *MIS-SEN* regulates endosperm development. We first investigated whether *MISSEN* functions in *cis* to regulate its nearby genes. Searching the genes 10 kb upstream and downstream of the *MISSEN* locus identified one putative protein-coding gene (LOC_Os08g35510) and three putative transposons (LOC_Os08g35520, LOC_Os08g35530, and LOC_Os08g35540) located near *MISSEN*. However, when we detected their expression levels in the T-DNA insertion mutant and the WT, no differences were found (Supplementary Fig. 4a), showing that *MISSEN* might not influence its nearby genes in *cis*.

It has been reported that lncRNAs could involve in the regulation of a number of cellular progresses[19,27], and the modes of actions are dependent on the subcellular localization of lncRNAs in cells[28]. Therefore, we firstly investigate the subcellular localization of *MISSEN*. We performed RNA fluorescence in situ hybridization (FISH) in *MISSEN*-OX roots with Cy3-labeled oligonucleotide probes specific to *MISSEN*[29]. Most of the fluorescent signals were observed in the cytoplasm of *MISSEN*-OX root cells that were hybridized with the *MISSEN* probe but not in the root cells of the WT plant (Fig. 3a). A weaker signal was also detected in nucleus (Fig. 3a). We have also separated nuclear and cytoplasmic fractions in caryopsis at 3DAP and *MISSEN* was detected to accumulate mostly in cytoplasm (Fig. 3b). These results suggested that *MISSEN* is mainly located in the cytoplasm and exerts its function there.

To search for potential *MISSEN*-interacting proteins, we next performed RNA pull-down assays with biotinylated *MISSEN*, the antisense sequence of *MISSEN* as control, followed by mass spectrometry (MS) (Fig. 3c). To purify the *MISSEN* RNA–protein complex, we incubated in vitro-transcribed *MISSEN* bound to beads with developing seed extracts and identified the pulled-down proteins by mass spectrometry (Fig. 3c, Supplementary Data 1). Among the highly enriched proteins, only a HeFP (LOC_Os03g38990) was detected from three independent RNA pull-down assays.

The interaction between *MISSEN* and HeFP was further confirmed by RNA immunoprecipitation (RIP) assay using the HeFP-GFP fusion protein in the rice protoplast system and tRNA-scaffolded Streptavidin Aptamer (tRSA)-RNA pull-down assay[30,31] using in vitro-transcribed tRSA-tagged *MISSEN*

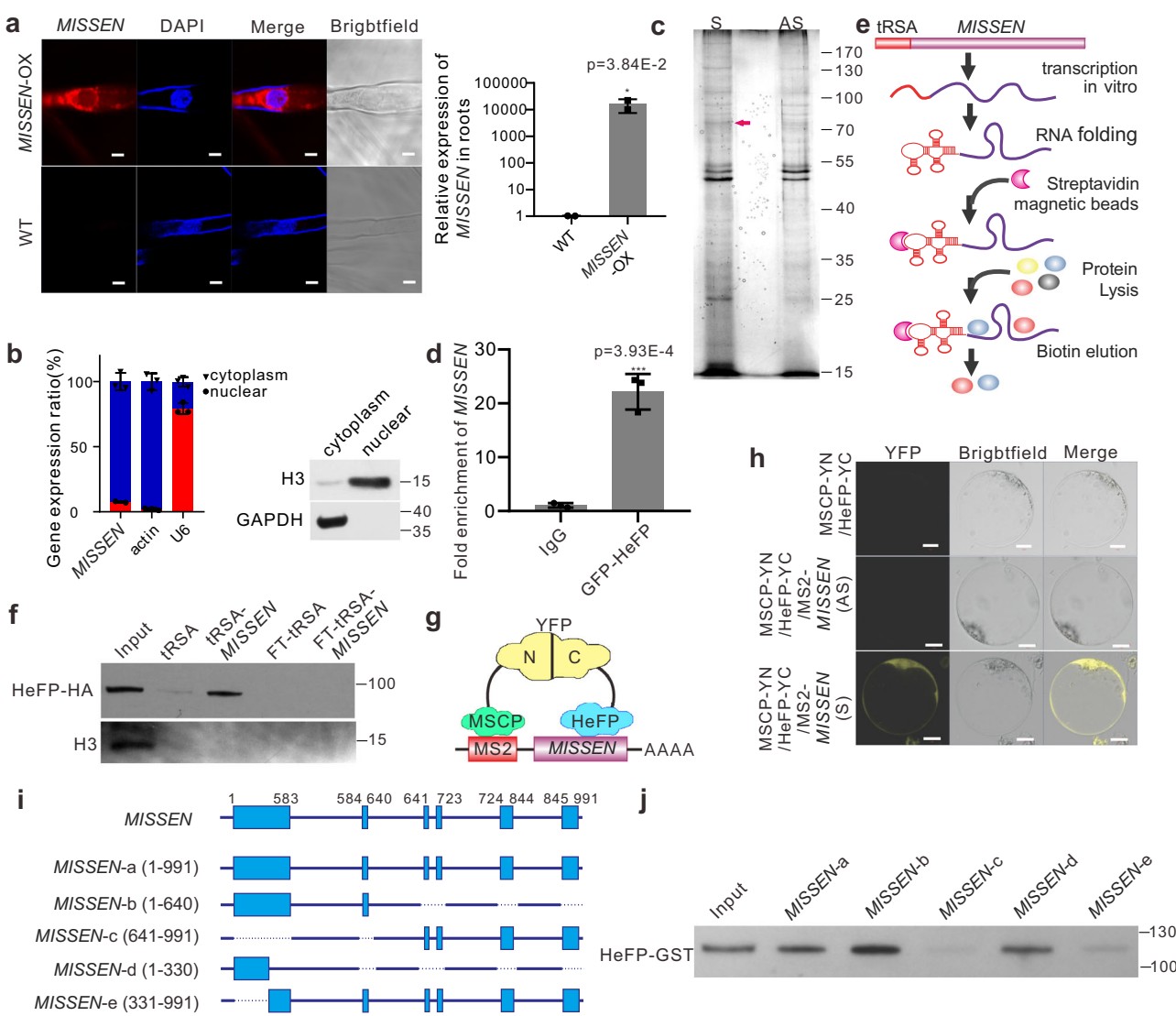

**Fig. 3 The lncRNA *MISSEN* interacts with HeFP. a** Subcellular localization of Cy3-labeled *MISSEN*. Scale bars, 5 μm. The expression level of *MISSEN* in roots of WT and *MISSEN*-OX plants. Values are the means ± SD ($n = 3$ replicates, normalized against *ACTIN2*). **b** Nuclear separation experiments with qRT-PCR analysis shows that *MISSEN* is predominantly localized in the cytoplasm. Values are the means ± SD ($n = 3$ replicates). **c** Silver staining of the *MISSEN* RNA pull-down assay. The red arrow indicates HeFP. **d–f** RNA immunoprecipitation assay (**d**) and tRSA-RNA pull-down assay (**e**, **f**) of *MISSEN* and HeFP. Values are the means ± SD ($n = 3$ replicates, normalized against *ACTIN2*). "FT" in **f** means "Flow through". The experiment was repeated three times with similar results and one representative result was shown. Schematic diagram of tRSA-RNA pull-down assay was showed in **e**. **g** Schematic diagram of the TriFC assay. **h** TriFC assay of *MISSEN* and HeFP. nYFP was fused to MSCP, and cYFP was fused to HeFP, 12xMS2 nucleotide sequences were fused to sense or antisense *MISSEN*. Scale bars, 10 μm. The experiment was repeated three times with similar results and one representative result was shown. **i** *MISSEN* fragments used for binding domain identification. **j** Identification of the HeFP-binding domain on *MISSEN* by RNA pull-down and western blot. The experiment was repeated three times with similar results and one representative result was shown. Significant differences were identified at the 5% (*), 1% (**), 0.1% (***) and <0.01% (****) probability levels using two-tailed paired *t*-tests.

(Fig. 3d–f). The schematic diagram for the tRSA-RNA pull-down assay method is shown in Fig. 3e. The results further verified the specificity of this interaction. We also checked the in vivo association of *MISSEN* transcripts with HeFP using a trimolecular fluorescence complementation (TriFC) assay using a MS2 system[32]. Figure 3g is schematic diagram. The fluorescent signals showed that sense *MISSEN* transcripts colocalized with HeFP in the cytoplasm but antisense *MISSEN* transcripts did not (Fig. 3h), indicating that *MISSEN* directly interacts with HeFP in the cytoplasm.

To find the exact interaction between *MISSEN* and HeFP, and understand the functional domains of the lncRNA, we divided the *MISSEN* transcript into three sections in accordance with the

lncRNA length and exonic components (Fig. 3i): *MISSEN*-a, the full-length sequence *MISSEN* (1–991 nucleotides (nt)), *MISSEN*-b, the first half of *MISSEN* (1–640 nt) including the first two exons, and *MISSEN*-c (641–991 nt), the rest of *MISSEN*. Then, the full-length sequence and the two truncated mutants were fused with tRSA, a tRNA scaffold that includes a streptavidin aptamer, for RNA pull-down assay[30,31]. RNA pull-down assay using in vitro purified HeFP-GST followed by western blot determined their interaction. The results showed that, beside the full length, *MISSEN*-b of the lncRNA has more ability to bind to HeFP compared with *MISSEN*-c (Fig. 3j), indicating that HeFP mainly binds to the first two exon of *MISSEN* (Fig. 3j). We further narrowed down *MISSEN*-b segment into *MISSEN*-d only with the

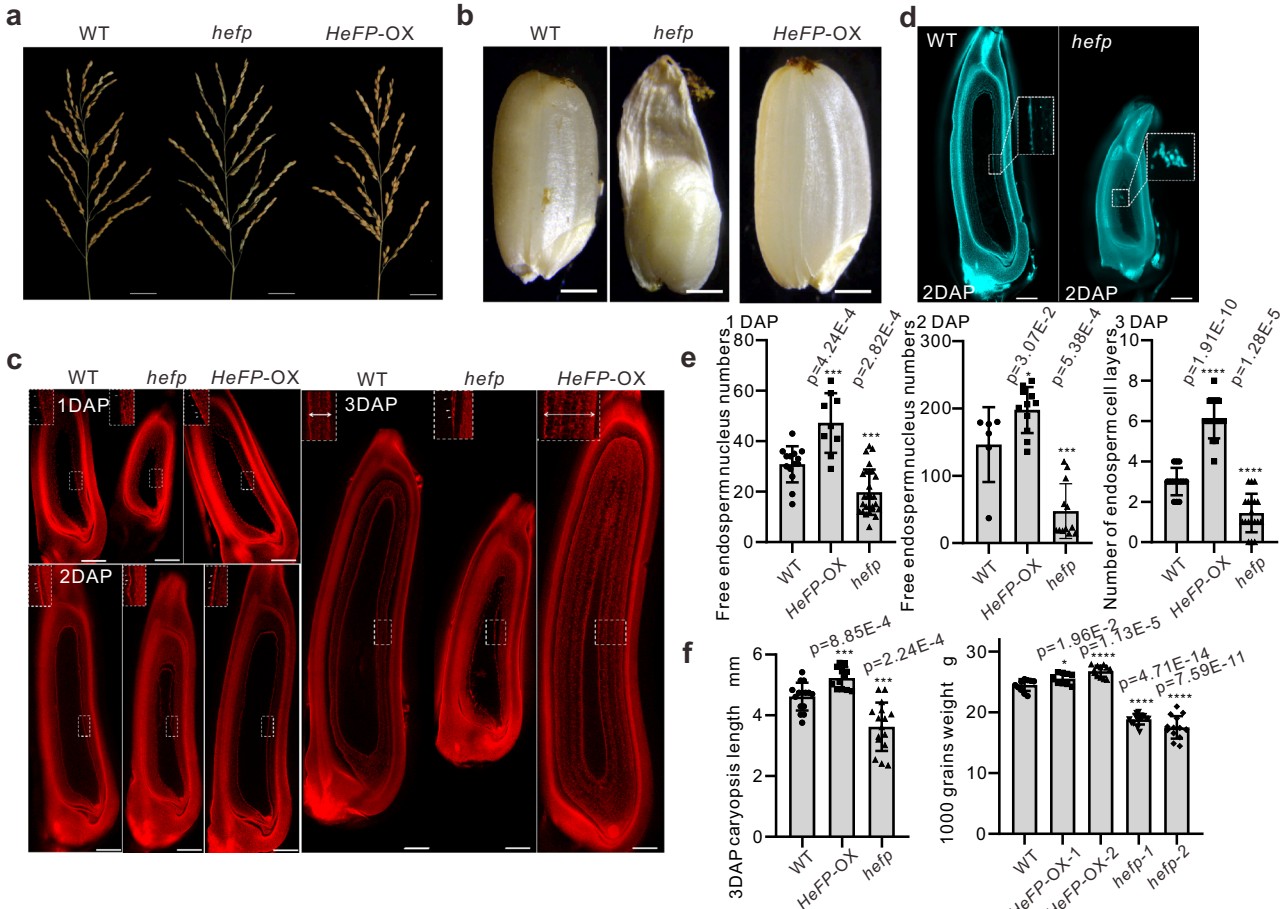

**Fig. 4 Phenotype analysis of WT, *hefp*, and *HeFP*-OX plants. a** The panicles of WT, *hefp*, and *HeFP*-OX plants. Scale bars, 3 cm. **b** The seeds of WT, *hefp*, and *HeFP*-OX plants. Scale bars, 1 mm. **c** Confocal laser scanning microscopy of developing caryopses stained with Eosin B in WT, *hefp*, and *HeFP*-OX endosperm. Scale bars, 50 μm. **d** Confocal micrograph of caryopses at 2 DAP stained with Propidium Iodide in WT and *hefp* caryopses. Scale bars, 200 μm. **e** The number of free endosperm nuclei in WT, *HeFP*-OX, and *hefp* plants at 1DAP ($n = 14, 9, 27$ caryopsis) and 2DAP ($n = 6, 11, 12$ caryopsis), and the number of endosperm cell layers in WT, *HeFP*-OX, and *hefp* plants at 3DAP ($n = 14, 15, 19$ caryopsis). Values are the means ± SD. **f** The caryopsis length of WT, *HeFP*-OX, and *hefp* plants at 3DAP ($n = 15, 12, 16$ caryopsis) and the 1000 grains weight of mature seeds in WT, *HeFP*-OX-1, *HeFP*-OX-2, *hefp*-1 and *hefp*-2 plants ($n = 11, 10, 12, 15, 14$ plants). Values are the means ± SD. Significant differences were identified at the 5% (*), 1% (**), 0.1% (***) and <0.01% (****) probability levels using two-tailed paired *t*-test.

first exon (1–330 nt), and the rest of *MISSEN* as *MISSEN*-e (331–991 nt) (Fig. 3i) and found that the first exon of *MISSEN* (*MISSEN*-d) is sufficient for the binding of HeFP (Fig. 3j). The results indicated that the first exon of the lncRNA is important for it to action.

**HeFP positively regulates endosperm development.** The observations above demonstrated the interaction between *MISSEN* and HeFP. HeFP is predicted to be a helicase family member. *HeFP* was expressed in most of tissues analyzed with a higher expression level in caryopsis (Supplementary Fig. 4b). The helicase family member ENL1 was shown to regulate syncytial endosperm development, but the effect of HeFP on endosperm is unclear. To determine if *MISSEN* affects endosperm development by regulating HeFP, we constructed *HeFP* knockout (*hefp*) and overexpression (*HeFP*-OX) transgenic plants (Supplementary Fig. 4c), and obtained a T-DNA insertion mutant of *HeFP*. T-DNA insertion on the exon of *HeFP* (Supplementary Fig. 4d) induced sterility in rice plants, but not affected vegetative growth (Fig. 4a, Supplementary Fig. 4e and f). Similar to the *MISSEN*-OX plants, the pollen grains and embryo sac of *hefp* plants were fertile and the pollen germinated normally (Supplementary Fig. 4g–i),

but their seeds were morphologically abnormal and the endosperm was not fully developed (approximately 12.7% and 17.6% of ovaries in two individual *hefp* plants developed normally after pollination, most of the *hefp* plants had abnormal endosperm), exhibited a " dent and bulge " shape or shriveled (Fig. 4b and Supplementary Fig. 4l), and similar to those of the *MISSEN-OX* (Fig. 1i), some exhibited more severe endosperm defects (Supplementary Fig. 4l). The T-*hefp* exhibited the similar phenotype with *MISSEN*-OX and *hefp* plants (Supplementary Fig. 4j and k). By observing whole-mount samples stained with Eosin B or PI, we found that the nuclei distribution and nuclei division was abnormal in *hefp* plants at 2 DAP (Fig. 4c–e), which was similar to that of *MISSEN*-OX plants (Fig. 2a–c). At 3 DAP, the *HeFP*-OX plants have several layers of cellularized endosperm cells (see magnified images) while the *hefp* endosperm only has one layer of endosperm nuclei (Fig. 4c and e). In particular, widths of the top half of the *hefp* caryopsis were obviously thinner compared with those of the WT at 3 DAP (Fig. 4c) and mature stage (Fig. 4b), and appears a dent and bulge in the seed shape. Moreover, the *HeFP*-OX plants had a slightly enlarger grains when compared with the WT plants (Fig. 4b and f). These results indicate that HeFP is required for endosperm development.

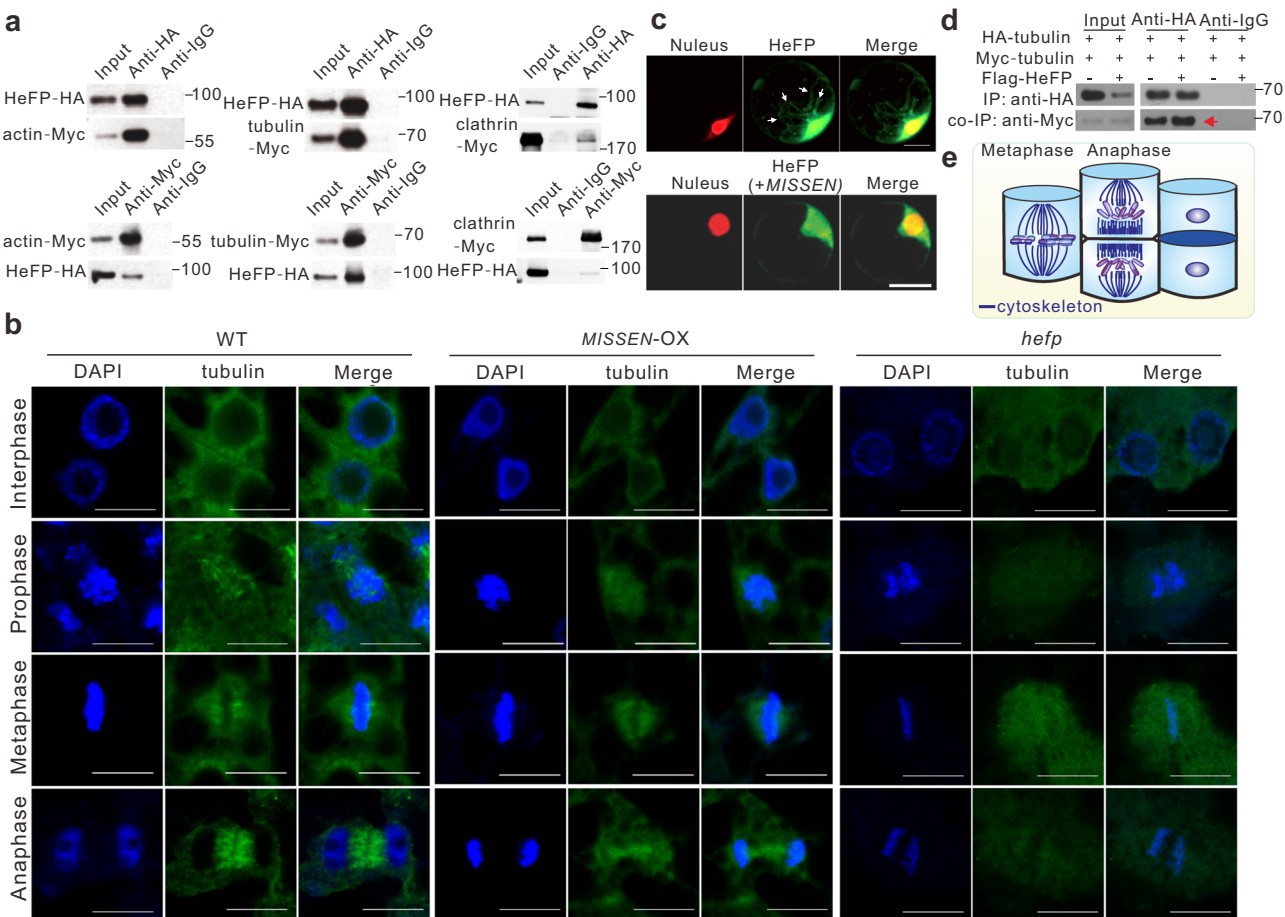

**Fig. 5 HeFP and *MISSEN* affect microtubule polymerization. a** In vitro binding assays between HeFP and actin, tubulin, and clathrin. The experiment was repeated three times with similar results and one representative result was shown. **b** Microtubule arrays were visualized by immunostaining with anti-alpha-tubulin during mitosis in WT, *MISSEN*-OX, and *hefp* endosperm cells at 3 DAP. Microtubules are colored green, and nuclei or chromosomes are colored blue. Scale bars, 10 μm. The experiment was repeated three times with similar results and one representative result was shown. **c** Subcellular localization of HeFP with or without *MISSEN*. Arrows indicate the filamentous distribution of HeFP. Scale bars, 10 μm. The experiment was repeated three times with similar results and one representative result was shown. **d** Co-express HA-tagged and Myc-tagged tubulin in protoplast and evaluate the level of microtubule polymerization by co-IP assay to detect the degree of interaction between HA-tagged and Myc-tagged tubulin. The result showed that HA-tagged tubulin immunoprecipitated more Myc-tagged tubulin when HeFP was overexpressed (indicate by an arrow), suggesting HeFP facilitates microtubule polymerization. The experiment was repeated three times with similar results and one representative result was shown. **e** Schematic diagram of the cytoskeleton during endosperm mitosis.

**MISSEN affects endosperm development by regulating HeFP, which in turn affects cytoskeletal proteins**. Plant genomes encode a large number of helicase proteins, and these helicases are reported to have diverse functions[33–35], but the role of HeFP during endosperm development has not been reported. To address how HeFP affects endosperm development, we next performed co-immunoprecipitation (co-IP) to identify proteins that bind to HeFP (Supplementary Fig. 5a, Supplementary Data 2). Interestingly, HeFP-interacting proteins were enriched in cellularization-related proteins, including tubulin, actin, clathrin, and formin. In vitro binding assays further confirmed the interaction between HeFP and these proteins except for formin (Fig. 5a, Supplementary Fig. 5b). The interaction between helicases and cytoskeletal proteins has also been reported in the INO80-family helicases in animals[36,37], suggesting that HeFP might function through interacting with cytoskeletal proteins.

Among the three HeFP-interacting proteins, actin and tubulin showed potential to participate in the endosperm development as both were reported to regulate nuclear movement[38,39]. Clathrin is an important component of vesicles and functions in cell plate formation[40], and thus might not be able to explain nucleus division and distribution referring to *MISSEN* and HeFP function. Therefore, we firstly performed the immunofluorescence experiments to investigate the effects of HeFP on actin and tubulin. As shown in Fig. 5b and Supplementary Fig. 5c, the effect of HeFP on actin is not significant as that of tubulin. As tubulin proteins are important for mitosis and cell division[41], and HeFP was shown to regulate nuclei division, distribution, and endosperm cellularization. We therefore asked whether HeFP could affect endosperm development by regulating tubulin function and cytoskeletal organization during endosperm development.

To address this hypothesis, we first examined the subcellular localization of HeFP and found that HeFP exhibited filamentous distribution, suggesting that HeFP might locate on cytoskeleton (Fig. 5c, upper panel). We then co-expressed both HA-tagged and Myc-tagged tubulin, together with or without HeFP in the rice protoplast system and found that HeFP binds to both α-tubulin and β-tubulin and promoted the polymerization of tubulin (Fig. 5d and Supplementary Fig. 5d). It is well known that a functional interzonal phragmoplast is formed between the separating sister nuclei, depositing a periclinal endosperm cell

wall that separates the outer cell layer from the original layer of alveoli. The same process of nuclear microtubules and cell wall formation are repeated until the endosperm is fully cellular[2], during which abnormal microtubular behavior may result in the abortion of endosperm[2]. Figure 5e is the schematic diagram, showing that the mitotic spindles in metaphase and anaphase in periclinal orientation in an endosperm alveolus. To address whether *MISSEN* and HeFP indeed affected microtubular behavior, we observed the microtubules of endosperm cells at 3 DAP in *hefp*, *MISSEN*-OX, and WT plants. We performed an immunofluorescence assay with antibodies against tubulin, which were in turn recognized by secondary antibodies (goat anti-mouse IgG (H + L) AF488). This assay showed that in WT endosperm at 3 DAP each free nucleus was surrounded by an interactive microtubule system at mitosis prophase; however, in *hefp* and *MISSEN*-OX transgenic plants, the green fluorescence signals were extremely weak and the microtubules exhibited an abnormal morphology (Fig. 5b). At metaphase, the microtubules assembled in WT endosperm cells, but in *hefp* and *MISSEN*-OX endosperm cells the microtubule signal was weak and irregular (Fig. 5b). At anaphase, the boundary between adjacent microtubule domains was formed in WT endosperm cells but not in *hefp* and *MISSEN*-OX endosperm cells (Fig. 5b).

Together, our results suggest that HeFP is essential for cytoskeleton polymerization during endosperm mitosis and cellularization. As both *HeFP* loss of function and *MISSEN* overexpression inhibit cytoskeletal polymerization, we proposed that high levels of *MISSEN* might block the function of HeFP.

**MISSEN competitively inhibited the interaction between HeFP and tubulin**. We have showed that the first exon of *MISSEN* (1–330 nt) specifically bound to HeFP (Fig. 3i and j). We next asked whether *MISSEN* competitively inhibits the interaction between HeFP and tubulin to block the function of HeFP in regulation of tubulin. To this end, we investigated the binding domains of HeFP interacted with *MISSEN* or tubulin. The HeFP protein was divided into three sections representing the T2SSE and RuvBN domains (HeFP2, 1–192 amino acids (aa)), the AAA11 domain (HeFP3, 193–420 aa), and the AAA12 domain (HeFP4, 421–651 aa) (Fig. 6a). Each section or the full-length HeFP protein (HeFP1, 1–651 aa) was fused with a HA tag and used for immunoprecipitation with *MISSEN* or tubulin. Immunoblot detection in rice protoplast cells transfected with different expression vectors showed that deletion of the AAA11 and AAA12 domains (193–651 aa) eliminated the interaction with *MISSEN* (Fig. 6b).

We further verified the interaction domains between *MISSEN* and HeFP by TriFC assays, including (1) truncated mutants of HeFP with or without *MISSEN* binding domain (deletion of AAA1 and AA12 domains); (2) truncated mutants of *MISSEN* with or without HeFP-binding sequences. The results showed that the AAA11 and AAA12 domains of HeFP and the first exon of *MISSEN* are required for the interaction between *MISSEN* and HeFP (Supplementary Fig. 6a and b). Interestingly, we found that the AAA11 and AAA12 domains are also the main binding site of tubulin (Fig. 6c), implying that *MISSEN* and tubulin may compete to bind HeFP.

To validate the observations above, we co-expressed Myc-tagged tubulin and HA-tagged HeFP in rice protoplast cells. When co-expressing *MISSEN*, the interaction between tubulin and HeFP was significantly impaired (Fig. 6d, upper panel). We further used *HeFP*-OX protoplast cells which spontaneously express HeFP with a GFP tag, and detected the interaction of HeFP with the endogenous tubulin using anti-tubulin when transfected *MISSEN* or not. The result showed that *MISSEN*

indeed inhibited the interaction between endogenous HeFP and tubulin (Fig. 6d, down panel). As the above results (Fig. 5c, upper panel) showed that HeFP might locate on cytoskeleton, we next investigate the effect of *MISSEN* on the interaction between HeFP and tubulin. We co-expressed *MISSEN*, the filamentous distribution of HeFP in the cytoplasm disappeared (Fig. 5c, down panel), implying that *MISSEN* negatively regulated the binding of HeFP to tubulin. It can be concluded that that *MISSEN* competitively inhibited the interaction between HeFP and tubulin, which may in turn affect cytoskeleton polymerization during endosperm mitosis and cellularization. To further demonstrate the downstream pathways *MISSEN* involving in, we analyzed the differentially expressed genes (DEGs) between *MISSEN*-RNAi and WT plants through transcriptome (Supplementary Fig. 7a). The results showed that the DEGs enriched in cell wall regulatory, carbohydrate synthesis and storage pathways (Supplementary Fig. 7b and Supplementary Data 3 and 4), which further supported the roles of *MISSEN* on cellularization and endosperm development. The Schematic diagram that *MISSEN* inhibited the interaction between HeFP and tubulin was shown in Fig. 6e.

**MISSEN is a parent-of-origin lncRNA and is precisely controlled by H3K27me3 levels during endosperm development**. Finally, we questioned how *MISSEN* is precisely controlled during seed development. We detected its expression patterns during ovule development, including before and after pollination as well as during seed development. In the WT, *MISSEN* was present at high levels in the pistil on the day before pollination, and its levels decreased quickly after pollination, especially at 3 DAP (Fig. 7a), indicating that precise control of *MISSEN* is important for endosperm development.

Histone modifications within promoter regions have important functions in regulating gene expression. These dynamic and reversible modifications affect chromatin accessibility to control the transcriptional status of genes. H3K27me3 is a common histone modification in promoter regions and acts to repress transcription[42]. EMF2a is an important component of polycomb repressive complex 2 (PRC2) which participates in the modification of H3K27me3, and we reanalyzed the public data to analyze the level of H3K27me3 in WT and in *emf2a* mutants on the promoter region of *MISSEN*[43]. H3K27me3 modification was detected in the −2.5 kb upstream of *MISSEN* and was down-regulated in the *emf2a* mutant (Supplementary Fig. 7c). We then performed chromatin immunoprecipitation (ChIP) to analyze the pattern of H3K27me3 (−1 kb upstream to 0.5 kb downstream) on *MISSEN* during different stages of seed development. In the mature WT pistils, the H3K27me3 modification was low in the region from −279 to −45 bp upstream of the 5′ terminal of *MISSEN*. However, at 3 and 4 DAP, the H3K27me3 modification was increased in the −279 to −95 bp region and in the −244 to −45 bp region (Fig. 7b). A negative association was observed between the histone modification level and *MISSEN* levels after pollination. We also obtained the *emf2a* mutant and applied RT-qPCR to detect the expression of *MISSEN*. The results showed that *MISSEN* was up-regulated in the *emf2a* mutant (Supplementary Fig. 7d). These results suggested that *MISSEN* is transcriptionally regulated by H3K27me3 during endosperm development.

The expression pattern of *MISSEN* implied that it might be a parent-of-origin lncRNA. Then we examined whether *MISSEN* is precisely regulated in a parent-of-origin-specific manner and is critical for normal growth and development. To address this question, we edited the 181st nucleotide at the first exon of *MISSEN* by clustered regularly interspaced short palindromic repeats (CRISPR)/CRISPR-associated protein 9 (Cas9) genome

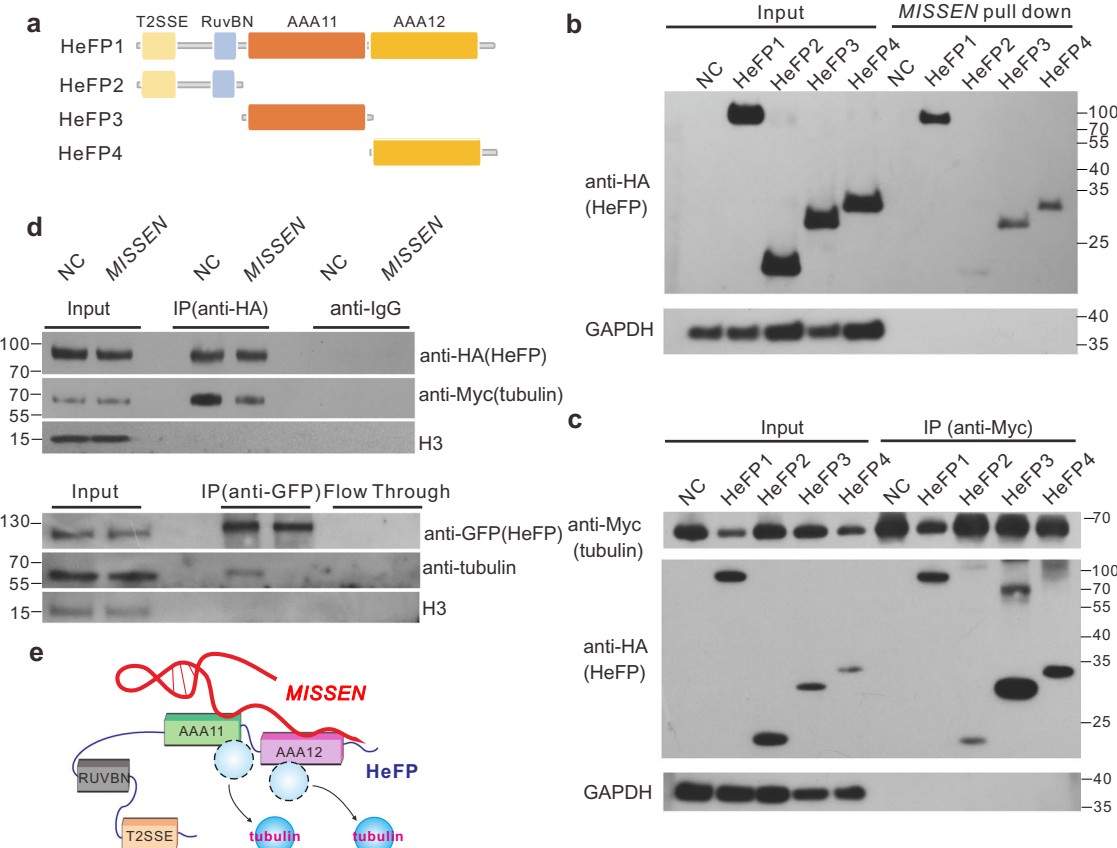

**Fig. 6 MISSEN competitively inhibited the interaction between HeFP and tubulin. a** HeFP sections used for binding domain identification. **b** Identification of the MISSEN binding domain by a MISSEN pull-down assay. The experiment was repeated three times with similar results and one representative result was shown. **c** Identification of the tubulin binding domain by immunoprecipitation. The experiment was repeated three times with similar results and one representative result was shown. **d** In vitro competitive binding assay of MISSEN in rice protoplast cells generated from WT (up panel) and HeFP-OX (down panel, detecting inherent tubulin) seedlings. The experiment was repeated three times with similar results and one representative result was shown. **e** The schematic diagram that MISSEN inhibited the interaction between HeFP and tubulin.

editing (generating a new MISSEN-edited line (MISSEN[ed.] line)). Then we crossed the MISSEN[ed.] transgenic plants with WT plants (using the MISSEN[ed.] line as the male or female parent) and examined the MISSEN transcripts in the endosperm (Fig. 7c). The result showed that MISSEN is a maternally expressed gene because the edited MISSEN transcripts were expressed in the endosperm only when using the MISSEN[ed.] line as the female parent (Fig. 7d).

Our data indicated that the parent-of-origin lncRNA MISSEN suppresses nucleus division, distribution and endosperm cellularization by blocking the function of HeFP, thereby impairing cytoskeletal polymerization during endosperm development. Figure 8 is the working model of MISSEN during endosperm development.

**Discussion**

Endosperm development includes two important processes at early stages, an initial syncytial phase followed by a cellularization phase. Cellularization is critical for endosperm development, and the correct onset of endosperm cellularization could determine the seed size and even seed viability[44,45]. Therefore, understanding the molecular mechanisms of endosperm cellularization is important. However, in rice, only a few mutant-based molecular studies of early endosperm development have been conducted; therefore, our knowledge regarding the molecular mechanisms underlying endosperm development in cereals is lacking. In this study, we discovered that a maternally expressed

cytoplasmic lncRNA MISSEN is essential for rice endosperm development. We further demonstrated that MISSEN functions through hijacking the helicase protein HeFP, leading to impaired cytoskeletal polymerization during endosperm development. This lncRNA therefore has potential applications in rice breeding.

LncRNAs are essential modulators of a wide range of biological processes and function through diverse mechanisms in a variety of eukaryotes[19,27]. MISSEN might be a decoy lncRNA, which can sequester proteins from their targets of action. In plants, several lncRNAs have similar functional mechanisms to that of MISSEN. For example, the lncRNA ALTERNATIVE SPLICING COMPETITIVE FACTOR (ASCO) binds NSR splicing factors and competes with the alternative splicing targets of NSRs[46], and lncRNA ENOD40 binds to NSRs and regulates their relocalization from nuclear speckles to cytoplasm[47]. This mechanism of lncRNAs hijacking proteins or altering their subcellular localization facilitates the rapid fine-tuning of protein activity. As a maternally expressed cytoplasm-localized lncRNA, MISSEN expression is controlled by H3K27me3 and MISSEN negatively regulates the function of HeFP, ultimately suppressing endosperm development. A group of maternally and paternally expressed lncRNAs was identified during endosperm development, but their functions have remained largely uninvestigated. Our study proposes a model for how maternally expressed lncRNAs affect post-embryonic development.

Another contribution of the study is revealing the novel function of the HeFP. Plant genomes encode a large number of

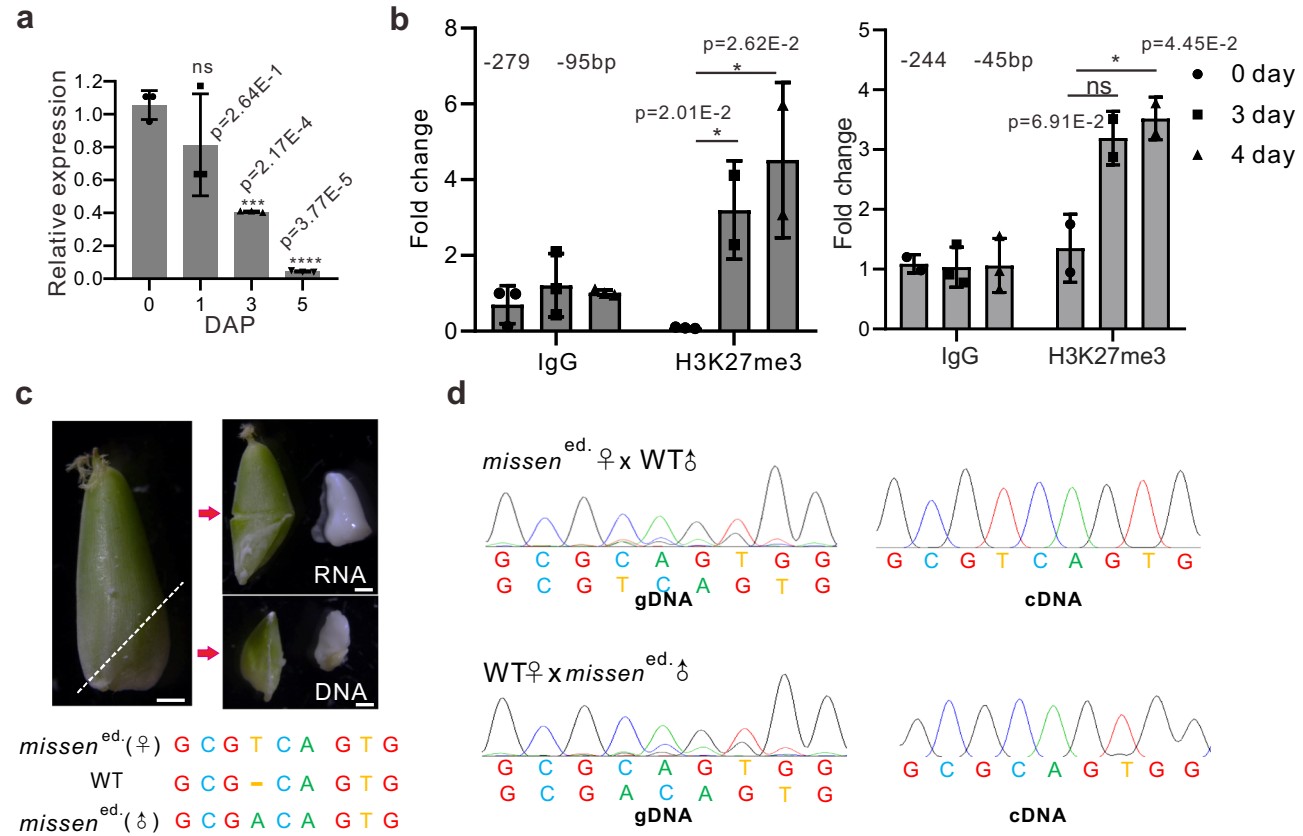

**Fig. 7 Parent-of-origin analysis of the lncRNA _MISSEN_. a** The expression pattern of _MISSEN_ in developing WT caryopses. Values are the means ± SD (_n_ = 3 replicates, normalized against _ACTIN2_). **b** ChIP analysis of the H3K27me3 modifications in the promoter region of _MISSEN_. Values are the means ± SD (_n_ = 3 replicates, normalized against IgG). **c** Schematic diagram of endosperm separation. Scale bars, 1 mm. **d** The genotypes and expressed _MISSEN_ transcripts in endosperm of the hybrid offspring of _MISSEN_ed. transgenic plants with WT plants. Significant differences were identified at the 5% (*), 1% (**), 0.1% (***) and <0.01% (****) probability levels using two-tailed paired _t_-test.

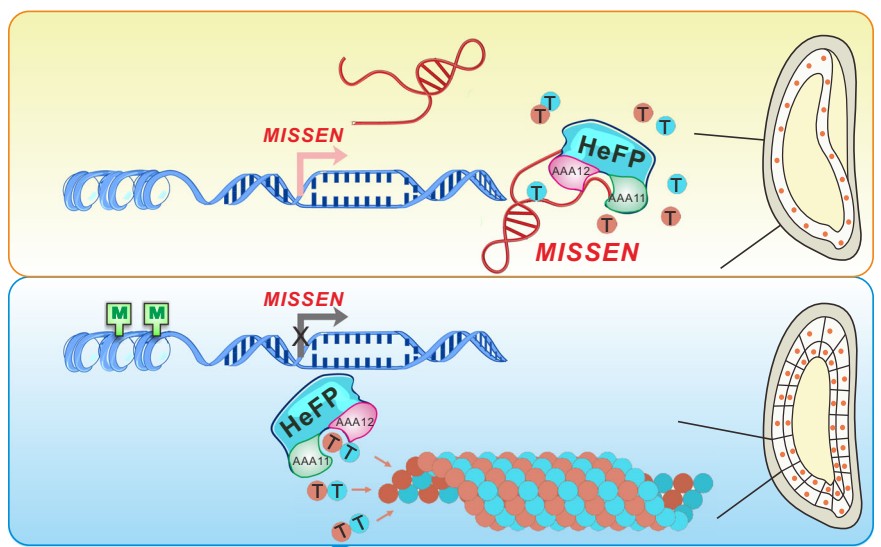

**Fig. 8 The working model of _MISSEN_ during endosperm development.** _MISSEN_ competitively inhibited the interaction between HeFP and tubulin. The transcription of _MISSEN_ is turned off by H3K27me3 modification during endosperm development.

helicase proteins, and these helicases are reported to have diverse functions[34,35]. Until now, only one member of the helicase family member ENL1 was shown to regulate endosperm development in rice, and ENL1 is mainly involved in syncytial stage of endosperm[14], chromosomal segregation failure was observed in

ENL1 loss-of-function mutants, causes nuclear division abnormally. In this study, we found that HeFP is essential for endosperm development. HeFP affect microtube behavior, resulting in abnormal cytoskeletal polymerization during endosperm development. The interaction between helicases and cytoskeletal

proteins has also been reported in the INO80-family helicases in animals[37]. It can be concluded that the functions of HeFP are related to either microtubule functions or chromosome dynamics that are important for endosperm development. It was also noted that cytoskeletal proteins, especially those microtubule related proteins were recently revealed to play important roles in crop agronomic traits, including rice grain yield[48,49], together with the finding in this study, we can prospect that further studies on the helicase family would provide novel insight into the understanding of rice grain development.

Seed yield and quality are two of the most important traits in crop improvement. Importantly, the *MISSEN*–HeFP pathway has a potential use for grain quality improvement, such as improving the embryo to endosperm ratio. As an essential regulator of endosperm development, HeFP is difficult to manipulate because knockout of *HeFP* causes very serious endosperm defects. Instead, modifying the expression level and pattern of *MISSEN* by overexpression or by modulating the H3K27me3 histone modification on the *MISSEN* locus to fine tune the activity of HeFP during postembryonic development will offer more practical approach. Thus, *MISSEN* might be important for the future genetic improvement of grain quality traits in rice.

## Methods

### Plant growth conditions, generation of transgenic rice plants, and phenotype analysis

The Zhonghua 11 (*O. sativa japonica*) rice cultivar was used for experiments. The growth conditions and generation of transgenic plants were conducted according to Zhang et al. [50]. Briefly rice seeds from the control plants and the transgenic plants imbibed in darkness for 2 d at 30 °C and then were grown for ~20 d in a soil seed bed at 28 °C, 70% humidity (12 h light/12 h dark), and then the seedlings were transplanted to a field in Guangzhou, China (23°08′N, 113°18′E), where the growing season extends from late April to late September. The mean minimum temperature range was 22.9–25.5 °C, and the mean maximum temperature range was 29.7–32.9 °C. The day length ranged from 12 to 13.5 h. Plants were cultivated using routine management practices.

The *MISSEN* T-DNA mutant is a enhancer trap lines obtained from Rice Mutant Database of Huazhong Agricultural University[51]. The *MISSEN* and *HeFP* knockout mutants were generated using CRISPR-Cas9-based genome editing technology[52]. The *MISSEN* and *HeFP* overexpression transgenic plants were constructed using pRGV vector. The pRNAi-35S binary vector, which has been described by Zhang et al. was used to generate the *MISSEN* RNAi mutant[53]. The *HeFP* T-DNA insertion mutants was collected from POSTECH, and the *emf2a* mutant was a gift from professor Chen of Agricultural College of Yangzhou University, Yangzhou, China[54,55].

The T$_2$ and T$_3$ generations of the transgenic plants were used for phenotypic analyses. The phenotypes in the T$_2$ and T$_3$ generations were stable. The primers are listed in supplementary Table 1.

### Plant material collection, RNA extraction, and qRT-PCR

The caryopsis from WT, *MISSEN*-OX, *MISSEN*-RNAi, *HeFP*-OX, and *hefp* plants were collected. Total RNA was extracted with TRIzol reagent (Invitrogen, CA, USA) and reverse-transcribed using the PrimeScriptTM RT reagent kit (Takara, Japan) according to the manufacturer's instructions. qRT-PCR was carried out using SYBR Premix Ex TaqTM (Takara, Japan). *ACTIN2* was used as reference genes. The RT-PCR was performed according to the manufacturer's instructions (Takara, Japan), and the resulting melting curves were visually inspected to ensure the specificity of the product detection. Quantification of gene expression was performed using the comparative Ct method. Experiments were performed in triplicate, and the results are represented as the mean ± standard deviation (SD). The primers are listed in supplementary Table 1.

### 5′ RACE and 3′RACE

Total RNA from WT spikelet was extracted using liquid nitrogen and TRIzol reagent (Invitrogen, CA, USA) according to the manufacturer's guidelines. The 5′- and 3′-ends of cDNA were acquired using a 5′-FULL RACE Kit with TAP (Takara, Japan) and 3′-FULL RACE Core Set with PrimeScript RTase (Takara, Japan), respectively, according to the manufacturer's instructions. PCR products were obtained and then cloned into pEASY-Blunt (TransGen Biotech, China) for further sequencing. The primers of 5′-FULL RACE are outer 5′-GGAATAAATAAGGCACAGGGAGAACAC-3′ and inner 5′-ACCGCAAGA-CAACCACCCTCA-3′. The primers of 3′-FULL RACE are outer 5′-TGCAGCC-GAGCTCACCGTAG-3′ and inner 5′-CTGAGGGTGGTTGTCTTGCGTCT-3′.

### Rice protoplast transient transformation

Two-week-old rice shoots were used for protoplast isolation. About 100 rice plants were cut into ~0.5-mm strips with

propulsive force using sharp razors. The strips were incubated in enzyme solution (1.5% cellulose RS, 0.75% macerozyme R-10, 0.6 M mannitol, 10 mM MES, pH 5.7, 10 mM CaCl$_2$, and 0.1% BSA) for 4–5 h in the dark with gentle shaking. Following enzymatic digestion, an equal volume of W5 solution (154 mM NaCl, 125 mM CaCl$_2$, 5 mM KCl, and 2 mM MES, pH 5.7) was added, and samples were shaken for 30 min. Protoplasts were released by filtering through 40-μm nylon mesh into round-bottom tubes and were washed three to five times with W5 solution. The pellets were collected by centrifugation at 150×*g* for 5 min in a swinging bucket. After washing with W5 solution, the pellets were then resuspended in MMG solution (0.4 M mannitol, 15 mM MgCl$_2$ and 4 mM MES, pH 5.7) at a concentration of 2 × 10$^6$ cells mL$^{-1}$. Aliquots of protoplasts (200 μL) were transferred into a 2-mL round-bottom microcentrifuge tube and mixed gently with 20 μg plasmid DNA. Transfected protoplasts were collected by centrifugation for 5 min at 100×*g*, resuspended and then incubated at 28 °C in the dark for 20 h.

### TriFC assays

We modified BiFC vectors pUC19-YN and pUC19-YC to generate TriFC assays vectors with a MS2 system[32,56]. Briefly, MSCP was fused to pUC19-YN, *HeFP* was fused to pUC19-YC, 12×MS2-*MISSEN*-S or 12×MS2-*MISSEN*-AS was fused to pUC19-YN without the sequence of nYFP. PEG-mediated transfections were performed as previously described[57]. Briefly, Aliquots of protoplasts (200 μL) were transferred into a 2-mL round-bottom microcentrifuge tube and mixed gently with 20 μg plasmid DNA. Transfected protoplasts were collected by centrifugation for 5 min at 100×*g*, resuspended and then incubated at 28 °C in the dark for 20 h. Protoplasts were observed using a Zeiss7 DUO NLO LSM880 confocal laser microscope (Carl Zeiss, Germany) with excitation/emission wavelengths of 514/519–563 nm.

### Whole-mount developing caryopses observation

Flowering spikelets were markered at noon and collected thereafter. The collected ovary or caryopsis samples were fixed in FAA (50% ethanol:formalin:acetic = 89:5:6, v/v) in vacuo for 30 min. After replacing the solution with new FAA or 70% ethanol, the samples were incubated overnight. Samples were sequentially hydrated in 50% ethanol, 30% ethanol, and distilled water.

For the whole-mount eosin B-staining CLSM (WECLSM), the ovaries were pretreated in 2% aluminum potassium sulfate for 20 min. Then the ovaries or caryopsis were stained with 10 mg/L eosin B (C$_{20}$H$_6$N$_2$O$_9$Br$_2$Na$_2$, FW 624.1) solution (dissolved in 4% sucrose) for 16–20 h at room temperature. The samples were post-treated in 2% aluminum potassium sulfate for 20 min. Next, the samples were rinsed with distilled water three times, and dehydrated with a series of ethanol solutions (30%, 50%, 70%, 90%, and 100%). Then, samples were treated with a series of solutions consisting of absolute ethanol and methyl salicylate (4:1, 3:1, 2:1, 1:1, 1:2, 1:3, and 1:4) each for 4 h and cleared in pure methyl salicylate twice each for 4 h. Finally, samples were kept at 4 °C overnight in methyl salicylate.

As for PI-staining, the rehydrated caryopsis were washed twice with PBS (0.9% NaCl, 50 mM phosphate buffer, pH 7.4) for 30 min. After treatment with RNase A (100 μg/ml) at 37 °C overnight, the samples were stained with PI (5 μg/ml) at 4 °C overnight in the dark, washed with PBS three times each for 3 h, dehydrated by an ethanol series (30%, 50%, 70%, 80%, 90%, and 100%), and then washed twice with absolute ethanol for 30 min. Next, samples were treated with a series of solutions consisting of absolute ethanol and methyl salicylate (4:1, 3:1, 2:1, 1:1, 1:2, 1:3, and 1:4) each for 4 h and cleared in pure methyl salicylate twice each for 4 h. Finally, samples were kept at 4 °C overnight in methyl salicylate.

CLSM images were taken using the zeiss 7DUO NLO LSM880 laser confocal microscope (Carl Zeiss, Germany) with excitation/emission wavelengths of 561/566–630 and 561/566–610 nm for the PI-stained sample and eosin B observation, respectively.

### Paraffin section

Caryopsis were collected and fixed in FAA (50% ethanol:formalin:acetic = 89:5:6, v/v) and then vacuum-infiltrated for 30 min, replacing the solution with new FAA or 70% ethanol and incubating overnight. Dehydrated in an ethanol series (30%, 50%, 70%, 80%, 90%, and 100%) and then washed twice in absolute ethanol for 30 min. Embedded in melted Paraplast (Sigma, USA) and sectioned to a thickness of 8 μm with a microtome (Leica Microsystems RM2145). After dewaxing with TO transparent agent, the sections were hydrated through an ethanol series (100%, 90%, 70%, 80%, 50%, and 30%) and then washed twice in distilled water for 30 min, stained with 1% toluidine blue for 45 s, then sealed with resin. Sections were visualized under a Zeiss Axio Imager Z2 microscope (Carl Zeiss, Germany).

### Fluorescence immunohistochemistry analysis

For immunolocalization of tubulin in endosperm cells during mitosis, caryopsis of WT, *MISSEN*-OX, and *hefp* were used at 3 DAP. Caryopsis were fixed in 4% (w/v) paraformaldehyde in microtubule-stabilizing buffer/DMSO (MTSB: 50 mM PIPES, 5 mM EGTA, 5 mM MgCl$_2$, and 5% DMSO, pH 6.7–7.0) at room temperature under vacuum for 1 h and then overnight at 4 °C. Fixed caryopsis were dehydrated in an ethanol series (30%, 50%, 70%, 80%, 90%, and 100%) and then washed twice in absolute ethanol for 30 min. Embedded in melted Steedman's wax[58], and sectioned to a thickness of 8 μm with a microtome (Leica Microsystems RM2145). After dewaxing with ethanol, hydrated the sample with MTSB. The plasma membrane was

permeabilized with 1% (v/v) Triton X-100 in 10% (v/v) DMSO-MTSB for 30 min at room temperature. Blocking with 1% (w/v) BSA in MTSB for 1 h at room temperature, anti-alpha tubulin DM1A (Abcam, USA) was diluted 1:50 in 3% (w/v) BSA in MTSB and incubated overnight at 4 °C. After washing in MTSB for three times, anti-mouse Fluorescein isothiocyanate (FITC)-conjugated secondary antibody (Invitrogen, CA, USA) was used at 1:300 dilution in 3% (w/v) BSA in MTSB. DNA was stained with 1 mg/mL DAPI (Sigma, USA), endosperm cells were observed on a Zeiss7 DUO NLO LSM880 confocal laser microscope (Carl Zeiss, Germany).

**Chromatin immunoprecipitation assay**. One gram of pistil, caryopsis at 3 and 4 days after pollination were collected and crosslinked with 1% formaldehyde for 30 min in vacuo. Crosslinking was stopped with Gly at the final concentration of 0.125 M. The tissues were then ground to fine powder in liquid nitrogen, and ChIP experiments were done using the EpiQuik Plant ChIP Kit (Epigentek, USA). Briefly, the cell lysates were sonicated for DNA shearing and incubated with the antibody-bound (anti-Histone H3 trimethyl K27 antibody, Abcam, USA; anti-IgG, provided by the kit) assay plate for immunoprecipitation. The protein/DNA complex was cleaned and reverse-crosslinked. Following protease K digestion, DNA was purified by the column. Primers were designed to amplify the primer DNA region of *MISSEN*.

**Western blotting**. Proteins extracted from protoplast cells were resolved by 10% Bis–Trispolyacrylamide gels and were transferred to polyvinylidene fluoride (PVDF) membranes. Membranes were blocked in 5% BSA for 1 h and probed with the appropriate antibody overnight at 4 °C and then were incubated with horseradish peroxidase-conjugated secondary antibodies at room temperature for 1 h. Membranes were visualized with an enhanced chemoluminescence detection system. The antibodies used in this study were listed in Supplementary Table 2.

**Immunoprecipitation**. For IP assays, GFP- or HA-tagged HeFP, Myc- or HA-tagged tubulin, Myc-tagged actin, Myc-tagged clathrin, and Myc-tagged formin were used. To identify interaction proteins of HeFP-GFP (C-terminal) in spikelet, we used the anti-GFP (AlpaLife, China) for immunoprecipitation. In the Co-IP experiment with HeFP, tubulin, actin, clathrin and formin in protoplast cells, anti-HA (Sigma, USA), anti-mouse IgG (Thermo, USA), anti-rabbit IgG (CST, USA), anti-Myc (Proteintech, USA) and MagnaBind Protein G Beads (Thermo, USA) were used, and the samples were boiled after vigorous washing for three times. All proteins for IP were lysed with IP lysis buffer supplemented with Cocktail (Thermo, USA). Finally, all samples were suspended in loading buffer and then denatured for 5 min at 100 °C, separated via SDS–PAGE, transferred to PVDF membranes, and blotted. The information of antibodies used were described in supplementary Table 2.

**Purification of full-length HeFP**. cDNA fragment of *HeFP* was amplified by PCR from cDNA generated from spikelet total RNA using the following primers: forward 5′-CGGGATCCGCGGGGCGGAGTGGCG-3′ and reverse 5′-CGGAATTCGCTCTGGTATTCTGATGCACTCAAGTACT-3′. Then the cDNA fragment was cloned into a pET-N-GST-Thrombin-C-His vector containing an N-terminal fusion of a GST tag. Recombinant HeFP with termination codon in pET-N-GST-Thrombin-C-His was transformed into *E. coli* expression strain BL21 [Transetta(DE3)] chemically competent cell (Transgen biotech, CD801)] to express full-length HeFP. 5 mL Luria-Bertani (LB) medium supplemented with kanamycin was inoculated with a single colony at 37 °C. After overnight growth, the culture was diluted 100-fold into 100 mL LB culture supplemented with kanamycin for 5 h. The expression of HeFP protein was induced in the presence of 0.4 mM IPTG at 28 °C for 4 h. Then the cell pellets were collected at 10,000×*g*, 4 °C for 10 min and were suspended with lysis buffer (500 mM NaCl, 50 mM Tris, pH 8.0, 5% Glycerol, 0.5 mM 1,4-dithiothreitol (DTT) and 1×Protease Inhibitor (cOmplete EDTA free, Roche, Switzerland)). The cells were lysed for 30 min on ice, followed by sonication at 230 V with 4 s on/6 s off for 25–30 min on ice. After centrifugation at 21,000×*g*, 4 °C for 10 min, the supernatant was applied to a ProteinIso® GST Resin column (TransGen Biotech, China) and the column was washed three times with lysis buffer. The bound protein was eluted with elution buffer. The HeFP protein was pooled and buffer exchanged into storage buffer (200 mM NaCl, 30 mM Tris, pH 8.0, 1 mM Tris(2-carboxyethyl)phosphine (TCEP), 20% glycerol) using an Amicon Ultra spin concentrator (Millipore, Germany). The purified HeFP was concentrated to about 0.3 mg/ml and stored at −80 °C. The quality of purified HeFP was further determined by SDS–PAGE.

**RNA immunoprecipitation**. In the RIP experiment, anti-GFP and anti-IgG antibodies were used along with an EZ-Magna RIP™ RNA-Binding Protein Immunoprecipitation Kit (17-701) (Millipore, Germany) according to the manufacturer's instructions. Protoplast cells were transfected with HeFP expression plasmid and *MISSEN* expression plasmid and collected after 16 h. All proteins for RIP were lysed with cell lysis buffer supplemented with cocktail (Thermo, USA) and Rnase inhibitor (Roche, Switzerland). 50 µL Protein A/G magnetic beads were incubated with 2 µg anti-GFP antibody or control IgG in 500 µL wash buffer at 4 °C for 1 h. Then the beads were washed three times and mixed with the cell lysates in new

tubes. The tubes were rotated at 4 °C overnight. Finally, RNA extraction from the beads was further collected by using TRizol according to the manufacturer's instructions. Reverse transcription and qPCR were performed as previously described.

**RNA pull-down assay**. *MISSEN* full-length sense and antisense sequence was cloned into pEASY-Blunt plasmid and its internal sequences were cloned into pEASY-Blunt plasmid containing the 5′ terminal tRSA tag. The plasmids were used as templates to in vitro transcribe RNA products using the TranscriptAid T7 High Yield Transcription Kit (Thermo, USA). The sense and antisense *MISSEN* RNAs were further labeled with biotin using Pierce Magnetic RNA-Protein Pull-Down Kit (Thermo, USA). Then the RNA products were purified using the GeneJET RNA Purification Kit (Thermo, USA). 50 pmol RNA per reaction system were denatured 5 min at 85 °C and slowly cooled down to room temperature. The powder of spikelets 3 days after pollination or protoplast cells transfected with HeFP expression vectors were harvested and resuspended with appropriate amount of IP lysis buffer with Rnase inhibitor (Roche, Switzerland) and Cocktail (Thermo, USA). The RNA pulldown assay was performed using the Pierce Magnetic RNA-Protein Pull-Down Kit in accordance with the manufacturer's instructions (Thermo, USA). The interacted proteins were determined by Mass spectrometry or Western blotting.

**Silver stainning and mass spectrometry**. Equal amounts of the retrieved proteins obtained from RNA pull-down assay were loaded on 10% SDS–PAGE gel. The gel was further stained using Pierce Silver Stain for Mass Spectrometry kit (Thermo Fisher Scientific) in accordance with the manufacturer's instructions. The specific bands were cut for mass spectrometry analysis.

**RNA in situ hybridization and immunofluorescence**. Cy3-labeled oligonucleotide probes specifically targeting *MISSEN* were designed and synthesized by RiboBio corporation (China). Root hair cells of WT and *MISSEN*-OX were collected fixed with 4% paraformaldehyde for 30 min and then permeabilized with 0.5% Triton X-100 for 30 min. Cells were incubated with Cy3-labeled FISH probes dissolved by 50% formamide in 2×SCC at 37 °C overnight. After hybridization, the nuclei were counterstained with DAPI. Cells were observed on a Zeiss7 DUO NLO LSM880 confocal laser microscope (Carl Zeiss, Germany).

**RNA-seq and data analysis**. The RNA isolated from 7-DAF-old caryopsis of the WT and *MISSEN*-RNAi was used for RNA-seq. Three biological replicates for each sample were set; each replicate consisted of about 12 caryopses. The samples were submitted to the ANROAD sequencing company (Beijing) for library preparation and sequencing. For the analysis of transcriptome sequencing, all clean reads were aligned to the reference genome *O. sativa* sp. *japonica* cv. Nipponbare (RGAP 7) using STAR v2.7.5c. The aligned read counts were obtained from alignment files using featureCounts v2.0.1. Differentially expressed genes analysis was conducted using the R package DESeq2 v1.32.0. The Gene Ontology enrichment analysis of the differentially expressed genes was performed on the AgriGOv2 website (http://systemsbiology.cau.edu.cn/agriGOv2/). Heatmaps were ploted using the R package Pheatmap v1.0.12.

For chip-seq analysis, all clean reads were aligned to the reference genome *O. sativa* sp. *japonica* cv. Nipponbare (RGAP 7) using bowtie2 version 2.4.1. Multi-mapped reads and PCR duplicated reads were discarded using samtools versions 1.9. peaks were called by the callpeak function of MACS2 v2.1.2 with options [-broad –broad-cutoff 0.1 -g 3.7e8].

**Reporting summary**. Further information on research design is available in the Nature Research Reporting Summary linked to this article.

## Data availability

The transcriptome datasets are uploaded to the NCBI SRA database (SRA Accession No. PRJNA765401). Source data are provided with this paper.

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

## Acknowledgements

This research was supported by the National Natural Science Foundation of China (Nos. 91940301, U1901202, 31801082 and 32100473), and the grants from Guangdong Province (2019JC05N394), and grants from Sun Yat-Sen University (No. 20lgzd26).

## Author contributions

Y.-F.Z., Y.-C.Z., Y.-M.S. carried out the experiments and drafted the manuscript. Y.Y., M.-Q.L., Y.-W.Y., J.-P.L., Y.-Z.F., Z.Z., L.Y., R.-R.H., J.-H.H., Y.C. and Y.-W.L. carried out loss-of/gain-of mutant screening and functional experiments. Y.-Q.C. conceived of the study, and participated in its design and coordination and helped to draft the manuscript. All authors read and approved the final manuscript.

## Competing interests

The authors declare no competing interests.
