## [Peer Review File · Nature Communications]

The parent-of-origin lncRNA MISSEN regulates rice endosperm developmentREVIEWER COMMENTS

Reviewer #1 (Remarks to the Author):

The manuscript entitled “The imprinted lncRNA BEERBELLY regulates rice endosperm development and grain shape” reports an interesting story on rice endosperm development. Authors found a long noncoding RNA (lncRNA), named as BEERBELLY. They confirmed the critical role of BEERBELLY in endosperm development first and based on this finding they further investigated (1) its downstream elements of the regulatory pathway and (2) its upstream regulatory factors. In their efforts for the regulatory mechanism underlying, authors respectively found that histone H3 lysine 27 trimethylation (H3K27me3) modification, a helicase family protein (HeFP) and tubulin are related to BEERBELLY and involved in the regulatory pathway.

Among the described results in this manuscript, to me, the most noteworthy result is the finding of a parent-of-origin lncRNA involved in endosperm development. As far as what I understand, lncRNA imprinting has not yet been reported in endosperm. In addition, in the field of parent-of-origin gene study lncRNA has not yet been included. What are the parental effects via lncRNA will be of very interest. The finding actually brings new references to the field for the consideration of parental roles in seeds development.

Generally, the methodology is sound and the quality of the data meet the standards required for publication. The methods are provided in detail for readers. However, before this work can be accepted for publication the extensive improvement in manuscript writing and data analysis are required.

My major concern:

1. Authors seem eager to construct a long complete story with accumulated data, but for some important conclusions required details have been neglected. For example, the basic expression patterns of described lncRNA and genes during sexual reproduction are necessary for proper evaluation of this work.
2. About the imprinted lncRNA. The Fig 7A showing the expression pattern of BEERBELLY in developing WT caryopses clearly indicates that the highest expression level appears before pollination, then it drops soon and reaches almost the bottom at 5 days after pollination. This pattern is a typical character of gamete-delivered parent-of-origin lncRNA, but not a character of imprinted lncRNA. Authors may understand that uniparental transcripts during endosperm development may arise due to gamete delivery during fertilization or genomic imprinting, thus, only detection of uniparental transcripts is not enough for confirming an imprinted lncRNA. According to the expression pattern shown in Fig 7A, most

likely BEERBELLY is not an imprinted lncRNA. Please carefully check it, but an egg cell-delivered maternal lncRNA.

3. About the BEERBELLY regulating Grain shape. From the relevant description in this manuscript we understand that the seed morphology is altered due to endosperm development defect. Since the most part of a rice seed is occupied by endosperm it is not surprising that once the endosperm development is arrested or blocked due to loss of function of some genes the seed morphology will accordingly modified, however, this is not what we usually mean regulating seed shape. It is just kind of phenotype actually. Interestingly, according to the description in this manuscript 35.6% aborted seeds of the beerbelly mutant showed a "beer belly" shape, but more aborted seeds (64.4%) exhibited even more severe phenotypic defects. Which is the typical phenotype?

4. About the BEERBELLY regulating cellularization. Authors pointed out that "BEERBELLY... negatively regulates endosperm cellularization". Also, authors described that "The defects in BEERBELLY-OX endosperm development started at the syncytial stage...the nuclei distribution and number were abnormal in BEERBELLY-OX plants... some nuclei were suspended in the middle of the embryo sac. At 2 DAP, the irregular distribution was more obvious and a prominent dent was found in the right upper of embryo sac in BEERBELLY-OX plants. ... At 3 DAP, the BEERBELLY-OX endosperm only has one layer of cellularized endosperm cells; walled compartments were irregularly formed". Clearly, the endosperm cellularization initiated normally in both BEERBELLY-OX and BEERBELLY-RNAi lines. Before endosperm cellularization the nuclei distribution and number were abnormal in BEERBELLY-OX plants. Since the irregular distribution of the nuclei will surely disturb the subsequent cellularization and a smaller number of the nuclei will result in less layers of the cellularized endosperm, we can suppose that BEERBELLY may regulate nucleus division and distribution via HeFP- tubulin pathway, but not really regulate the cellularization itself. The idea is supported by the evidences from BEERBELLY-RNAi lines as we see in lines the endosperm cellularization looks perfectly normal and only the layers of cellularized endosperm increased upon the nucleus regularly distributed.

Minor concerns:

1. In result 1, the authors concluded that there was no abnormality in beerbelly at the reproductive stage, which seems a premature conclusion. since firstly, the images of embryo sacs are not clear enough to estimate the development of embryo sac member cells and the corresponding statistics are lacking. Next, the I2-KI staining of pollen seems normal, but in vitro pollen germination and statistic analysis are needed to further verify the function of pollen. Embryo development should be carefully observed in BEERBELLY-RNAi, beerbellyed., BEERBELLY-OX plants that have slightly smaller embryo sacs, and the description of hefp and HeFP-OX reproductive development also need these data.

2. In RT-qPCR, the material used to extract RNA is spikelet, which includes different stages of flower before and after fertilization. better to use early seeds or endosperm to carry out relevant experiments.

3. 21.21% of the BEERBELLY-RNAi pistils had three stigmas, which implies an off-target effect in the RNAi plants. Do BEERBELLY-RNAi-1 and -2 come from different constructs or are they different lines from the same construct? In either case, more lines are needed for phenotypic analysis. In addition, hefp-1 and -2

resulted from editing the same target site, thus other target sites and knockout lines may help to exclude off-target effects and to understand the gene function.

4. Except for the seed development, the mutants and OX plants of BEERBELLY and HeFP appear no significant abnormal phenotype. Are BEERBELLY and HeFP specifically expressed during seed development?

5. Fig 1I, S1D, S3A, and S3C show that the mutant and OX embryos also have serious defects. Furthermore, the germination of the embryos in BEERBELLY-OX seeds was obviously abnormal (Fig S3C). Therefore, the authors need to add sufficient data on embryonic development for properly evaluating the major function of BEERBELLY.

6. In result 2, the author indicated that “the BEERBELLY-RNAi seeds grew faster and were slightly larger than those of the WT”; “the nuclei distribution and number were abnormal in BEERBELLY-OX plants”. In result 4, the author indicated that “most of the hefp plants had abnormal endosperm”; “width of the top half of the hefp caryopsis were obviously thinner compared with those of the WT at 3 DAP (Fig. 4C) and mature stage (Fig. 4B)”; “the HeFP-OX plants had a slightly enlarger grains”. For these results, the specific parameters or quantitative indicators, such as length, width and weight of seeds and number of free nuclei, must be statistically analysed instead of simply vague statements.

7. It is more logical that “identification of the HeFP binding domain on BEERBELLY (a part of Result 4)” should be combined with Result 3. Then, result 4 rephrase as “HeFP positively regulates endosperm development”.

8. In Result 5, authors indicate that HeFP can interact with tubulin, actin and clathrin. The subsequent experiments only explained how the HeFP dysfunction affects tubulin function, cytoskeletal depolarization and endosperm cellularization. The most obvious phenotype of hefp is that the endosperm is just formed at the micropylar end, i.e. HeFP affects the endosperm free nuclear migration and distribution at the syncytia stage. Many previous studies have verified that actin contributes to regulate nuclear movement. Does HeFP-actin/clathrin interaction actually play a role in nuclear migration? The detailed observation is required.

9. The H3 in Figure 6D is ambiguous.

10. Moreover, ChIP only proved that BEERBELLY may be repressed by H3K27me3 after pollination. There is also a possibility that H3K27me3 inhibited simultaneously the transcription of biparental genes, not just maternal or paternal genes, because the expression level of BEERBELLY is decreased and BEERBELLY should be cleared after fertilization to protect HeFP function.

11. In addition to ENL1, some other genes have also been reported to control the rice early endosperm development. for a comprehensive understanding of the mechanism of early endosperm development of rice, the introduction should be improved.

12. Statistical significance of Fig 1B, 1D, 1E, 3A, 3C, 7A, S2A, S2B and S4B are not shown in the figure legends, and Fig 7B is incomplete. Fig 1G does not show standard deviation and statistical significance.

13. The quality of the images in Fig 1A is poor, which may be caused by dehydration treatment of the sample is not sufficient or improper focus when taken pictures. The wild-type in Fig 1C is obviously

abnormal, showing an irregular distribution and cell layer of endosperm cells. Thus, the better pictures need to be provided.

14. The higher magnification images in Fig 2A, 4C and 4D are not sharp enough to accentuate the details and needs to be rearranged.

15. In result 2, "The top half of the BEERBELLY-OX ovaries.... At 7DAP, the BEERBELLY-OX endosperm... (Fig. S3B)." The information provided by the images does not support the author's description and conclusions.

16. Please indicate the full name of FT-tRSA in Fig 3E.

17. The statement of methods is too messy and needs to be integrated and rewritten in a better logicity. Such as, CLSM observation of early-stage endosperms can combine with eosin B staining, namely, whole-mount eosin B-staining CLSM (WECLSM). Plasmid Construction can be incorporated into the corresponding experiment.

18. In Fig 5A, IgG should be rephrased as Anti-IgG

19. The binding site of BEERBELLY and HeFP should be consistent in Fig 6A and Fig 8.

20. In Fig 8, tubulin is shown in two colors, which represents the two types of tubulin? This is not reflected in the results. Please explain.

Reviewer #2 (Remarks to the Author):

In their manuscript, Zhou et al., studied the role of a new long non coding RNA (lncRNA) called BEERBELLY in rice endosperm development. Authors found that BEERBELLY is a paternally imprinted lncRNA expressed in endosperm, and it negatively regulates endosperm cellularization. Interestingly, authors proposed that BEERBELLY functions through the hijacking of a helicase family protein (HeFP) to regulate tubulin function during endosperm cellularization. Authors also proposed that the expression of BEERBELLY is inhibited by histone H3 lysine 27 trimethylation (H3K27me3) modification after pollination. Overall this study is interesting and should be for a general audience. However, at the current state, there are several weak points, which I list below.

1. What is the molecule overexpressed in BEERBELLY in the T-DNA lines? Since the T-DNA is in the intron this needs to be better described. Is really similar to the Oex lines?

2. Fig 1H, the authors claimed that differences between RNAi and WT are "evident"? Please indicate arrows or what to look for non-rice experts? IT looks more than developmental phenotype, it is a growth phenotype. Hence, a kinetics of grain growth with standard deviations to compare RNAi lines and WT will better support the phenotype? Just looking to one grain in a Figure seems not enough. Furthermore, the authors mentioned a CRISPRcas9 line deleting the promoter of BEERBELLY in Fig. 7? Can these plants

be phenotyped for endosperm cellularisation? RNAi knock-down experiments are very good but having a clean loss of function mutant with CRISPr is possible now.

3. The RNAi phenotype is much less evident than the Oex. Can this further described at molecular level (e.g. endosperm transcriptome? Does HeFP expression is affected without BEERBELLY?).

4. Authors proposed that BEERBELLY is mainly located in the cytoplasm and exerts its function there. But this conclusion come from the RNA fluorescence in situ hybridization (FISH) done in BEERBELLY-OX using root cells which could lead to artefactual observations due to the over-expression in a different tissue. To be convincing authors must do it in WT condition and in the seed. Alternatively, separate nuclear vs cytoplasmic fractions in rice grains and test for lncRNA accumulation.

5. The authors have made interesting biochemical experiments to validate interactions however some controls are missing. For the TriFC assay, deletion mutants of MSHeFP without lncRNA binding (deletion of AAA1 and AA12 domains?) should be tested to conclude about the interaction. IN this assay, again all molecules are overexpressed. For example, different stabilities of sense and antisense sequences can also explain the differences observed (no binding on the antisense). Also test the truncated isoforms of BEERBELLY in the TriFC assays?

6. Fig. 5B is not easy to detect what the authors want to highlight. Does the lncRNA provoke any change in localisation? Better explain the “cytoskeleton” link in this figure concerning Ox BEERBELLY and “the cytoskeleton-like filar distribution of HeFP in the cytoplasm was impaired”. Idem in Fig. 5C, indicate in the figures what is the observation on the cytoskeleton?

7. Fig. 6D is capital to support the main conclusion on competition of HeFP. How many replicates of this experiment? Can this competition be observed in vivo on the Oex lines (preventing the binding of HeFP) to tubulin?

8. Authors suggested that BEERBELLY is transcriptionally regulated by H3K27me3 during endosperm development. This is supported only by correlative analysis of ChIP-QPCR. To be more convincing I suggest the authors (i) to use public data bases to analyze the level of H3K27me3 in WT and in different polycomb mutants on this promoter and (ii) if one mutant displays a difference on BEERBELLY locus to do an RT-QPCR to check its transcription level.

REVIEWER COMMENTS

Reviewer #1 (Remarks to the Author):

The manuscript entitled “The imprinted lncRNA BEERBELLY regulates rice endosperm development and grain shape” reports an interesting story on rice endosperm development. Authors found a Long noncoding RNAs (lncRNAs), named as BEERBELLY. They confirmed the critical role of BEERBELLY in endosperm development first and based on this finding they further investigated (1) its downstream elements of the regulatory pathway and (2) its upstream regulatory factors. In their efforts for the regulatory mechanism underlying, authors respectively found that histone H3 lysine 27 trimethylation (H3K27me3) modification, a helicase family protein (HeFP) and tubulin are related to BEERBELLY and involved in the regulatory pathway.

Among the described results in this manuscript, to me, the most noteworthy result is the finding of a parent-of-origin lncRNA involved in endosperm development. As far as what I understand, lncRNA imprinting has not yet reported in endosperm. In addition, in the field of parent-of-origin gene study lncRNA has not yet been included. What are the parental effects via lncRNA will be of very interest. The finding actually brings new references to the field for the consideration of parental roles in seeds development.

Generally, the methodology is sound and the quality of the data meet the standards required for publication. The methods are provided in detail for readers. However, before this work can be accepted for publication the extensive improvement in manuscript writing and data analysis are required.

Reply: We thank you for the comments and all the suggestions. Please see below our point-by-point responses.

My major concern:

Comment 1. Authors seems eager to construct a long complete story with accumulate data, but for some important conclusions required details have been neglected. For example, the basic expression patterns of described lncRNA and genes during sexual reproduction are necessary for proper evaluation of this work.

Reply: We agree and have performed additional experiments to analyze the expression patterns of *BEERBELLY* and *HeFP* in different tissues. The result showed that *BEERBELLY* was highly expressed in pistil and *HeFP* was expressed in most of tissues analyzed with a higher expression in caryopsis (please see **Fig. 1** below). The new data has been added to the revised manuscript and the revised **Figure S2B and S4B** (page 5, lines 143-144; page 9, lines 259-260, marked in blue), we appreciate the comments.

Figure 1. The expression pattern of *BEERBELLY* and *HeFP* in diverse tissues. **A.** The expression pattern of *BEERBELLY* in different tissues ($n = 3$ replicates, normalized against *ACTIN2*); **B.** The expression pattern of *HeFP* in diverse tissues ($n = 3$ replicates, normalized against *ACTIN2*).

Comment 2. About the imprinted lncRNA. The Fig 7A showing the expression pattern of *BEERBELLY* in developing WT caryopses clearly indicates that the highest expression level appears before pollination, then it drops soon and reach to almost the

bottom at 5 day after pollination. This pattern is a typical character of gamete-delivered parent-of-origin lncRNA, but not a character of imprinted lncRNA. Authors may understand that uniparental transcripts during endosperm development may arise due to gamete delivery during fertilization or genomic imprinting, thus, only detection of uniparental transcripts is not enough for confirming an imprinted lncRNA. According to the expression patten shown in Fig 7A, most likely BEERBELLY is not an imprinted lncRNA. Please carefully check it, but an egg cell-delivered maternal lncRNA.

Reply: We appreciate the comments and agree with the suggestion. Following the suggestion, we have redefined *BEERBELLY* as a parent-of-origin lncRNA and rephrased the description throughout the manuscript including title and main text (**page 1, line 2; page 2, line 38; page 13, line 374; page 14, lines 400 and 410** marked in blue). Thank you for the comment to make the concept more accurately.

Comment 3. About the BEERBELLY regulating Grain shape. From the relevant description in this manuscript we understand that the seed morphology is altered due to endosperm development defect. Since the most part of a rice seed is occupied by endosperm it is not superising that once the endosperm development is arrested or blocked due to loss of function of some genes the seed morphology will accordingly modified, however, this is not what we usually mean regulating seed shape. It is just kind of phenotype actually. Interestingly, according to the description in this manuscript 35.6% aborted seeds of the beerbelly mutant showed a "beer belly" shape, but more aborted seeds (64.4%) exhibited even more severe phenotypic defects. Which is the typical phenotype?

Reply: We agree and have described “grain shape” as a seed defect phenotype as suggested. We have rephrased the description throughout the manuscript including in title and main text (**page 1, lines 2 and 11; page 2, lines 44-45; page 5, line 130**). Regarding to the phenotype description, we apologized not clearly presenting the data.

What we originally want to express were: only 30.6% of the mature seeds were normally developed, 69.4% of the ovaries developed abnormally after pollination and had different morphologies. Among those abnormally developed (69.4%), 51.3% (35.6% of total) exhibited a typical "beer belly" shape (see below Fig. 2A, middle panel) and 48.7% (33.8% of total) exhibited even more severe phenotypic defects with less endosperm, but still showing "beer belly" shape (see below Fig. 2A right panel). For the concern, we have rephrased the description to make it clearer (page 5, lines 127-130, revised Figure 1D). Thank you.

Figure 2. The seed morphology and statistical ratio in different mutants. **A.** The morphology of seeds with a “beer belly” shape or less-endosperm. Scale bars, 1mm; **B.** The relative ratio of seeds with different morphologies.

Comment 4. About the BEERBELLY regulating cellularization. Authors pointed out that “BEERBELLY... negatively regulates endosperm cellularization”. Also, authors described that “The defects in BEERBELLY-OX endosperm development started at the syncytial stage...the nuclei distribution and number were abnormal in BEERBELLY-OX plants... some nuclei were suspended in the middle of the embryo sac. At 2 DAP, the irregular distribution was more obvious and a prominent dent was found in the right upper of embryo sac in BEERBELLY-OX plants. ... At 3 DAP, the BEERBELLY-OX endosperm only has one layer of cellularized endosperm cells; walled compartments were irregularly formed”. Clearly, the endosperm cellularization initiated normally in both BEERBELLY-OX and BEERBELLY-RNAi lines. Before endosperm cellularization the nuclei distribution and number were abnormal in

BEERBELLY-OX plants. Since the irregular distribution of the nuclei will surely disturb the subsequent cellularization and a smaller number of the nuclei will result in less layers of the cellularized endosperm, we can suppose that BEERBELLY may regulate nucleus division and distribution via HeFP- tubulin pathway, but not really regulate the cellularization itself. The idea is supported by the evidences from BEERBELLY-RNAi lines as we see in lines the endosperm cellularization looks perfectly normal and only the layers of cellularized endosperm increased upon the nucleus regularly distributed.

Reply: Thank you for the comments. We focused on cellularization process in previous version because of the observations that, in *hefp* and *BEERBELLY-OX* plants, when we examined the microtubules of endosperm cells, the most obvious phenotype is that the boundary between adjacent microtubule domains was not formed at anaphase which indicated failure of cellularization (revised **Figure 5B**). Considering your suggestion, we have statistically analyzed the nucleus numbers in different transgenic plants during endosperm development and found that indeed the nucleus division and distribution was also irregulated in *BEERBELLY-OX* and *hefp* (revised **Figure 2B-C** and **4D-E**). Thus, we have taken the suggestion to add the description of “nucleus division and distribution” to the revised manuscript (**page 1, line 10; page 7, lines 199-200; page 14, lines 410-411**, marked in blue).

Minor concerns:

Comment 5. In result 1, the authors concluded that there was no abnormality in beerbelly at the reproductive stage, which seems a premature conclusion. since firstly, the images of embryo sacs are not clear enough to estimate the development of embryo sac member cells and the corresponding statistics are lacking. Next, the I2-KI staining of pollen seems normal, but in vitro pollen germination and statistic analysis are needed to further verify the function of pollen. Embryo development should be carefully observed in BEERBELLY-RNAi, beerbellyed., BEERBELLY-OX plants that have slightly smaller embryo sacs, and the description of *hefp* and HeFP-OX

reproductive development also need these data.

Reply: We understand the reviewer’s concern. To address this issue, following the suggestion, we have redone the experiment to observe the developing Caryopses. We have also performed statistical analysis of embryo sac sizes in different mutants including *BEERBELLY*-RNAi, *BEERBELLY*-OX, *hefp* and *HeFP*-OX and *in vitro* pollen germination analysis. Our new data showed that the pollen germination and embryo sac development were normal in these transgenic plants although the embryo sacs of *BEERBELLY*-RNAi plants were slightly enlarged compared with that of WT plants. All the mutants have an eight-nucleate embryo sacs and normal embryo at 7 DAP (see **Fig. 3 below**). We have added the data to the revised manuscript (**page 5, lines 152-153; page 9, lines 266-267**, marked in blue) and the revised **Figure S2E-H and S4G-I**. For the concern, we have rephrased the description “no difference” to “no apparent difference” to make it more accurately (**page 4, line 122**). Thank you for the suggestion.

Figure 3. The statistics and functional analysis of the embryo sacs and pollen from different mutants. **A.** The width and length of the embryo sacs in WT and different mutants; **B-D.** The morphology of the embryo sacs in WT and different mutants. Scale bars, 100 μ m; **E.** *In vitro* pollen germination in WT and different mutants. Scale bars, 100 μ m; **F.** The morphology of the embryo in WT and different mutants at 7DAP. Scale bars, 100 μ m. The numbers in the images (B-D) indicate the proportion of samples that exhibited the phenotype.

Comment 6. In RT-qPCR, the material used to extract RNA is spikelet, which includes different stages of flower before and after fertilization. better to use early seeds or endosperm to carry out relevant experiments.

Reply: We agree and have performed additional RT-qPCR (see below **Fig. 4**) using the RNA extracted from endosperm at 3DAP or 5DAP. The relevant graphs were replaced (**page 17, line 487**, the revised **Figure S2C and S4C**). Thank you.

Figure 4. The relative expression of genes in endosperm in different mutants. **A.** The expression level of *BEERBELLY* in *BEERBELLY-OX* endosperm at 5DAP ($n = 3$ replicates, normalized against *ACTIN2*); **B.** The expression level of *BEERBELLY* in *BEERBELLY-RNAi* endosperm at 3DAP ($n = 3$ replicates, normalized against *ACTIN2*); **C.** The expression level of *HeFP* in *HeFP-OX* endosperm at 3DAP ($n = 3$ replicates, normalized against *ACTIN2*).

Comment 7. 21.21% of the *BEERBELLY-RNAi* pistils had three stigmas, which implies an off-target effect in the RNAi plants. Do *BEERBELLY-RNAi-1* and -2 come from different constructs or are they different lines from the same construct? In either case, more lines are needed for phenotypic analysis. In addition, *hefp-1* and -2 resulted from editing the same target site, thus other target sites and knockout lines may help to exclude off-target effects and to understand the gene function.

Reply: For the concern, we have analyzed the stigma numbers using additional 3 *BEERBELLY-RNAi* lines which came from the same construct but with different expression levels of *BEERBELLY*. The results showed that all the lines have a certain number of pistils with three stigmas ($11.4 \pm 6.8\%$) compared with that of WT plants, and the ratio positively correlated with the knock down efficiency of *BEERBELLY* (see Fig. 5A below). Thus “three stigma” might not be off-target effect. However, we currently cannot explain the mechanism regarding to the effects of the lncRNA on stigma number. We have added the data to the revised manuscript (page 5, lines 153-154 and revised Fig. S2J).

We have also added another mutant which has a T-DNA insertion on the 12th exon

of *HeFP* and analyzed its phenotypes (see below Fig. 5B-F). Consistently, T-DNA insertion on *HeFP* caused similar phenotypes with that of the *hefp-1* and *hefp-2* plants, which further supported the functions of *HeFP* on endosperm development. We have added the data to the revised manuscript (page 9, lines 264-265, 272-273, marked in blue) and revised Figure S4D, S4F and S4J-K accordingly. Thank you for the suggestion.

Figure 5. Phenotypic analysis of *BEERBELLY*-RNAi and T DNA insertion mutant *T-hefp*. **A.** Correlation analysis between RNAi interference level and three stigma ratio; **B.** The gene structure of the *HeFP*. The T-DNA insertion site of the mutant is indicated by the red arrow; **C.** Phenotypes of *T-hefp* plants. Scale bars, 15cm; **D.** The panicle of *T-hefp*. Scale bars, 3cm; **E.** The seeds of *T-hefp*. Scale bars, 1mm; **F.** The ratio of abnormal seeds in *T-hefp* plants($n = 13$ plants).

Comment 8. Except for the seed development, the mutants and OX plants of

BEERBELLY and HeFP appear no significant abnormal phenotype. Are BEERBELLY and HeFP specifically expressed during seed development?

Reply:For the concern, we have analyzed the expression patterns of *BEERBELLY* and *HeFP* in different tissues according to your comment (comment 1). *BEERBELLY* was very highly expressed in pistil and *HeFP* was expressed in most of tissues analyzed with a higher expression in caryopsis. Thus *BEERBELLY* might play a role in pistil and HeFP might mainly regulate endosperm development. *HeFP* belongs to a big helicase family, thus *HeFP* mutants have no significant abnormal phenotype in other development stages, which might be caused by gene redundancy. The new data has been added to the revised manuscript (**page 5, lines 143-144; page 9, lines 259-260**)and the revised **Figure S2B and S4B**, Thank you.

Comment 9. Fig 1I, S1D, S3A, and S3C show that the mutant and OX embryos also have serious defects. Furthermore, the germination of the embryos in BEERBELLY-OX seeds was obviously abnormal (Fig S3C). Therefore, the authors need to add sufficient data on embryonic development for properly evaluating the major function of BEERBELLY.

Reply: Following the suggestion, we have performed additional experiments to observe embryonic development of *BEERBELLY*-OX plants. We collected samples at 5d, 7d after pollination and at mature stage. As shown in **below Fig. 6**, the embryos developed normally at 5DAP and 7DAP in *BEERBELLY*-OX plants. At mature stage, the embryo appeared normally with shoot, radicle, epiblast, scutellum and coleorhiza (see **below Fig. 6A**). In the germination process, the seeds in *BEERBELLY*-OX were grew slightly slow but they could develop into normal seedlings (see **Fig. 6B below**). We speculated that the slightly slow growth phenotype in germination of *BEERBELLY*-OX seeds might be caused by lack of endosperm, that embryo development depends on endosperm development¹. The new data has been added to the revised manuscript (**page 6, lines 177-178**, marked in blue) and the revised **Figure S3E-F**. Thank you.

Figure 6. The development and germination of embryo in WT and *BEERBELLY-OX* plants. **A.** The phenotype of embryo at 5DAP, 7DAP and mature stage in WT and *BEERBELLY-OX* plants. Scale bars of eosin B-staining and freehand section are 100 μ m and 1mm respectively. The numbers in the images indicate the proportion of samples that exhibited the phenotype; **B.** Germination of mature seeds of WT and *BEERBELLY-OX* plants. Scale bars, 1cm.

Comment 10. In result 2, the author indicated that “the *BEERBELLY-RNAi* seeds grew faster and were slightly larger than those of the WT”; “the nuclei distribution and number were abnormal in *BEERBELLY-OX* plants”. In result 4, the author indicated that “most of the *hefp* plants had abnormal endosperm”; “width of the top half of the *hefp* caryopsis were obviously thinner compared with those of the WT at 3 DAP (Fig. 4C) and mature stage (Fig. 4B)”; “the *HeFP-OX* plants had a slightly

enlarger grains”. For these results, the specific parameters or quantitative indicators, such as length, width and weight of seeds and number of free nuclei, must be statistically analysed instead of simply vague statements.

Reply: We agree. Following the suggestion, we have statistically analyzed the caryopsis size, seed weights and free nuclear numbers and cell layer numbers of endosperm in different transgenic plants (see **Fig. 7** below). The new data has been added into the revised manuscript (**page 6, lines 171-175; pages 6-7, lines 182-196; pages 9-10, lines 273-277, and page 10, lines 279-281, marked in blue**) and the revised **Figure 2B, 4E-F and S3C-D**, Thank you.

Figure 7. The statistics of phenotype in different mutant plants. **A.** The number of free endosperm nuclei in different mutants at 1 and 2DAP; **B.** the number of endosperm cell layers in different mutants at 3DAP; **C.** The 1000 grains weight of mature seeds in different mutants; **D.** The caryopsis length of different mutant plants at 3DAP.

Comment 11. It is more logical that “identification of the HeFP binding domain on BEERBELLY (a part of Result 4)” should be combined with Result 3. Then, result 4

rephrase as “HeFP positively regulates endosperm development”.

Reply: This is indeed a good suggestion. We have moved this part to result 3 as suggested in the revised manuscript (**pages 8-9, lines 241-255**, marked in blue), and the relevant images in original Figure 4E-F have also been moved into revised Figure 3 I-J, thank you.

Comment 12. In Result 5, authors indicate that HeFP can interact with tubulin, actin and clathrin. The subsequent experiments only explained how the HeFP dysfunction affects tubulin function, cytoskeletal depolarization and endosperm cellularization. The most obvious phenotype of *hefp* is that the endosperm is just formed at the micropylar end, i.e. HeFP affects the endosperm free nuclear migration and distribution at the syncytia stage. Many previous studies have verified that actin contributes to regulate nuclear movement. Does HeFP-actin/clathrin interaction actually play a role in nuclear migration? The detailed observation is required.

Reply: Indeed, HeFP can interact with tubulin, actin and clathrin in our protein-protein interactomes. Clathrin is an important component of vesicles and functions in cell plate formation², and thus might not able to explain nucleus division and distribution referring to *BEERBELLY* and HeFP function. As the reviewer mentioned, previous studies have reported that both actin and tubulin contribute to regulate nuclear movement. We previously focused only on tubulin not including actin were due to the following two reasons: (1) tubulin has been reported to play an important role on nuclear movement and division; (2) we previously have performed the immunofluorescence experiment data about the effect of HeFP on actin and tubulin, and found that the effect of HeFP on actin was not significant as that of tubulin (**below Figure 8 on actin and Figure 5B on tubulin**). Because no significant effect was observed, this data not shown in previous version. For the concern, we have provided the immunofluorescence experiment data about the effect of HeFP on actin (**below Figure 8**) and brief explanations in the revised manuscript for better

understanding the chosen of *BEERBELLY*- HeFP-tubulin axis (page 10, lines 301-302, marked in blue) and the revised **Figure S5C**. Thank you.

Figure 8. The effects of *BEERBELLY* and HeFP on HeFP-interacted proteins. Microfilament arrays were visualized by immunostaining with TRITC Phalloidin during mitosis in WT, *BEERBELLY*-OX and *hefp* endosperm cells at 3 DAP. Microfilaments are colored red, and nuclei or chromosomes are colored blue. Scale bars, 10 μ m;

Comment 13. The H3 in Figure 6D is ambiguous.

Reply: We have redone the experiment and improved the image (see the below **Fig. 9**), thank you. The revised image has been replaced the original one (revised **Figure 6D, the up panel**).

Figure 9. *In vitro* competitive binding assay of *BEERBELLY* in rice protoplast cells.

Comment 14. Moreover, ChIP only proved that *BEERBELLY* may be repressed by H3K27me3 after pollination. There is also a possibility that H3K27me3 inhibited simultaneously the transcription of biparental genes, not just maternal or paternal genes, because the expression level of *BEERBELLY* is decreased and *BEERBELLY* should be cleared after fertilization to protect HeFP function.

Reply: We agree and have taken your suggestion to use “parent-of-origin “in the revised manuscript (please see our reply to your comment 2), very good comment indeed, thank you again.

Comment 15. In addition to *ENL1*, some other genes have also been reported to control the rice early endosperm development. for a comprehensive understanding of the mechanism of early endosperm development of rice, the introduction should be improved.

Reply: Thank you for the comment. We have revised the introduction and added the relevant information and references accordingly (**page 3, lines 81-86**, marked in blue).

Comment 16. Statistical significance of Fig 1B, 1D, 1E, 3A, 3C, 7A, S2A, S2B and S4B are not shown in the figure legends, and Fig 7B is incomplete. Fig 1G does not show standard deviation and statistical significance.

Reply: We have provided the statistical analysis as suggested, please see the revised **Figure 1D and F**), the revised **Figure 3D**, the revised **Figure 4C**, the revised Figures 7A and 7B, the revised **Figure S2C** and S4B, and all the other statistical data as suggested, thank you.

Comment 17. The quality of the images in Fig 1A is poor, which may be caused by dehydration treatment of the sample is not sufficient or improper focus when taken pictures. The wild-type in Fig 1C is obviously abnormal, showing an irregular distribution and cell layer of endosperm cells. Thus, the better pictures need to be provided.

Reply: The reviewer means Fig 4A and C? According to the comment, we have redone the experiments (see **below Fig. 10**) and improved the quality of these images (the revised **Figure 4A and 4C**). Thank you.

Figure 10. The new images used to replace the original ones. **A.** The panicles of *hefp* and *HeFP-OX*. Scale bars, 3cm; **B.** The endosperm cell layer in WT at 3DAP. Scale bars, 100 μ m.

Comment 18. The higher magnification images in Fig 2A, 4C and 4D are not sharp enough to accentuate the details and needs to be rearranged.

Reply: We have improved the revised **Figure 2A, 4C and 4D**, thank you.

Comment 19. In result 2, “The top half of the BEERBELLY-OX ovaries.... At 7DAP, the BEERBELLY-OX endosperm... (Fig. S3B).” The information provided by the images does not support the author's description and conclusions.

Reply: According to the comment, we have rephrased the description (**page 6, lines 171-175**) and added statistic of caryopsis size in WT and *BEERBELLY*-RNAi plants after pollination (the revised **Figure S3C**). In addition, a dot-line was added in the revised **Figure 1 H** for highlighting the caryopsis differences among different mutants. Thank you.

Comment 20. Please indicate the full name of FT-tRSA in Fig 3E.

Reply: FT-tRSA in Fig. 3E was “Flow through of tRSA”, we have indicated the full name in the figure legend, thank you.

Comment 21. The statement of methods is too messy and needs to be integrated and rewritten in a better logicity. Such as, CLSM observation of early-stage endosperms can combine with eosin B staining, namely, whole-mount eosin B-staining CLSM (WECLSM). Plasmid Construction can be incorporated into the corresponding experiment.

Reply: We apologized for not clearly presenting the method statement. As suggested, we have rewritten these sections to make them clear (**pages 18-19, lines 532-556; and page 21, lines 620-624**, marked in blue), thank you.

Comment 22. In Fig 5A, IgG should be rephrased as Anti-IgG

Reply: We have revised **Figure 5A** as suggested, thank you.

Comment 23. The binding site of BEERBELLY and HeFP should be consistent in Fig

6A and Fig 8.

Reply: Thank you for the suggestion, we have revised **Figure 8** to make it consistent.

Comment 24. In Fig 8, tubulin is shown in two colors, which represents the two types of tubulin? This is not reflected in the results. Please explain.

Reply: We have performed additional experiments which showed that HeFP binds to the two types of tubulin (see **Fig. 11 below**). The data has been added to the revised manuscript (**page 11, lines 309-311**, marked in blue) and the revised **Figure S5D**. Thank you again for all your suggestions.

Figure 12. *In vitro* binding assays between HeFP and α -tubulin, β -tubulin. **A.** HeFP is interacted with α -tubulin; **B.** HeFP is interacted with β -tubulin.

Reviewer #2 (Remarks to the Author):

In their manuscript, Zhou et al., studied the role of a new long non coding RNA (lncRNA) called BEERBELLY in rice endosperm development. Authors found that BEERBELLY is a paternally imprinted lncRNA expressed in endosperm, and it negatively regulates endosperm cellularization. Interestingly, authors proposed that BEERBELLY functions through the hijacking of a helicase family protein (HeFP) to regulate tubulin function during endosperm cellularization. Authors also proposed that the expression of BEERBELLY is inhibited by histone H3 lysine 27 trimethylation (H3K27me3) modification after pollination. Overall this study is interesting and

should be for a general audience. However, at the current state, there are several weak points, which I list below.

Comment 1. What is the molecule overexpressed in BEERBELLY in the T-DNA lines? Since the T-DNA is in the intron this needs to be better described. Is really similar to the Oex lines?

Reply: Thank you for the comments. The *BEERBELLY* T-DNA mutant is a enhancer trap lines obtained from Rice Mutant Database of Huazhong Agricultural University³. Thus the T-DNA insertion in the second intron of *BEERBELLY* promotes the transcription of *BEERBELLY*. We have added the information in the revised methods (page 16, lines 474-475, marked in blue), thank you.

Comment 2. Fig 1H, the authors claimed that differences between RNAi and WT are “evident”? Please indicate arrows or what to look for non-rice experts? IT looks more than developmental phenotype, it is a growth phenotype. Hence, a kinetics of grain growth with standard deviations to compare RNAi lines and WT will better support the phenotype? Just looking to one grain in a Figure seems not enough. Furthermore, the authors mentioned a CRISPRcas9 line deleting the promoter of BEERBELLY in Fig. 7? Can these plants be phenotyped for endosperm cellularisation? RNAi knock-down experiments are very good but having a clean loss of function mutant with CRISPr is possible now.

Reply: We agree. Following the suggestion, we have added a dot-line in **Figure 1H** to indicate the differences of caryopsis size between RNAi and WT, and rephrased “developmental phenotype” to “growth phenotype” (page 6, line 159, marked in blue). In addition, as suggested, we have also statistically analyzed the seed size of *BEERBELLY*-RNAi and WT plants during seed growth after pollination (the revised **Figure S2L**) (see **Fig. 11 below**), which showed that caryopsis *BEERBELLY*-RNAi is slightly bigger than that of WT.

The CRISPR/Cas9 edited lines of *BEERBELLY* were constructed for analyzing the parent origin of *BEERBELLY* during pollination, thus only 1 nucleotide (T insertion for *BEERBELLY*^{ed.} ♀ and A insertion for *BEERBELLY*^{ed.} ♂) was inserted in the first exon of *BEERBELLY* which might not affect the function of *BEERBELLY*. We have not obtained a line with deletion of the promoter of *BEERBELLY*. Thank you for your suggestion.

Figure 11. The changes of the caryopsis length during 1-4 days after pollination in WT and *BEERBELLY*-RNAi plants.

Comment 3. The RNAi phenotype is much less evident than the Oex. Can this further described at molecular level (e.g. endosperm transcriptome? Does HeFP expression is affected without *BEERBELLY*?).

Reply: We understand the reviewer's concern. As the reviewer's comment (comment 2, reviewer 2), the RNAi phenotype is growth phenotype and thus it might not show as evident as that of *BEERBELLY*-OX. However, when observing endosperm development in different time points especially at 3DAP, we can find that more endosperm cell layers in *BEERBELLY*-RNAi than that in WT (the revised **Figure 2A, 2B and 2E**). For the concern and to further confirm the phenotype of *BEERBELLY*-RNAi lines, we have statistically analyzed the free nuclear numbers and the endosperm cell layers of *BEERBELLY*-RNAi plants and WT plants during endosperm development at 1DAP, 2DAP and 3DAP (see **Fig. 13A and B below**). The results showed that more free endosperm nuclei and more endosperm cell layers at 3DAP in *BEERBELLY*-RNAi than that in WT, suggesting the endosperm in

BEERBELLY-RNAi plants grew faster than that of WT plants, showing an opposite phenotype with that of *BEERBELLY*-OX. These additional statistical data have been added into the revised manuscript (**page 7, lines 187-196** and the revised **Figure 2B**)

Regarding to the mechanism, we proposed that *BEERBELLY* negatively promotes endosperm development via affecting cytoskeleton polymerization during endosperm mitosis and cellularization. To further support this proposal, we have taken the suggestion from the reviewer and collected the caryopsis of *BEERBELLY*-RNAi and WT at 7DAP for transcriptome sequencing with three replicates. The sequencing data indeed showed that a number of cell wall-related and carbohydrate synthesis and storage-related genes are significantly up-regulated in *BEERBELLY*-RNAi compared to WT (see Fig. 13D below), confirming that *BEERBELLY*-RNAi promotes endosperm growth. We have added the results in the revised manuscript (**page 12, lines 366-371**, marked in blue) and the revised **Figure S7A-B**.

In addition, the *HeFP* expression was not affected by *BEERBELLY* which showed by both sequencing data and RT-qPCR (see Fig. 13E and F below). Thank you for your suggestion.

Figure 13. Knockdown of *BEERBELLY* with RNAi promotes endosperm development. **A.** The number of free endosperm nuclei in different mutants at 1 and 2DAP; **B.** the number of endosperm cell layers in different mutants at 3DAP; **C.** Heatmap of the expression of differentially expressed genes in caryopsis of *BEERBELLY*-RNAi compared to WT at 7DAP; **D.** The effects of *BEERBELLY*-RNAi on the expression of cell wall-related genes (up panel) and carbohydrate synthesis and storage-related genes (down panel); **E-F.** *BEERBELLY* does not affect *HeFP* expression detected by RNA-seq(**E**) and RT-qPCR($n = 3$ replicates, normalized against *ACTIN2*)(**F**).

Comment 4. Authors proposed that *BEERBELLY* is mainly located in the cytoplasm and exerts its function there. But this conclusion come from the RNA fluorescence in situ hybridization (FISH) done in *BEERBELLY*-OX using root cells which could lead to artefactual observations due to the over-expression in a different tissue. To be convincing authors must do it in WT condition and in the seed. Alternatively, separate nuclear vs cytoplasmic fractions in rice grains and test for lncRNA accumulation.

Reply: We agree and performed additional experiments by separating nuclear vs cytoplasmic fractions in rice caryopsis at 3 DAP to detect lncRNA accumulation. As shown in the revised **Figure 3B**, *BEERBELLY* is mainly located in the cytoplasm (see **Fig. 14 below**). Thank you.

Figure 14. Nuclear separation experiments with qRT-PCR analysis shows that *BEERBELLY* is predominantly localized in the cytoplasm ($n = 3$ replicates).

Comment 5. The authors have made interesting biochemical experiments to validate interactions however some controls are missing. For the TriFC assay, deletion mutants of MSHeFP without lncRNA binding (deletion of AAA1 and AA12 domains?) should be tested to conclude about the interaction. IN this assay, again all molecules are overexpressed. For example, different stabilities of sense and antisense sequences can also explain the differences observed (no binding on the antisense). Also test the

truncated isoforms of BEERBELLY in the TriFC assays?

Reply: Following the suggestion, we have performed additional TriFC assays via deletion mutants of HeFP without lncRNA binding, including (1) truncated mutants of HeFP with or without lncRNA binding (AAA1 and AA12 domains); (2) truncated mutants of *BEERBELLY* with or without HeFP binding sequences; (3) different controls including sense and antisense sequences of *BEERBELLY*. The results showed that the AAA11 and AAA12 domain of HeFP and the first exon of *BEERBELLY* are required for the interaction between *BEERBELLY* and HeFP (see **Fig. 15 below**). We have added the results in the revised manuscript (**page 12, lines 346-351**, marked in blue) and the revised **Figure S6**. Thank you for your suggestion.

Figure 15. TriFC assay of truncated *BEERBELLY* and HeFP mutants. **A.** *BEERBELLY* fragments used for binding domain identification; **B.** TriFC assay of *BEERBELLY* fragments and HeFP. The antisense sequences of *BEERBELLY* fragments as control. Scale bars, 20µm ;**C.** TriFC assay of *BEERBELLY* and truncated HeFP sections. The antisense sequences of *BEERBELLY* as control. Scale bars, 20µm; **D.** HeFP sections used for binding domain identification.

Comment 6. Fig. 5B is not easy to detect what the authors want to highlight. Does the lncRNA provoke any change in localisation? Better explain the “cytoskeleton” link in this figure concerning Ox *BEERBELLY* and “the cytoskeleton-like filar distribution of HeFP in the cytoplasm was impaired”. Idem in Fig. 5C, indicate in the figures what is the observation on the cytoskeleton?

Reply: We apology not clearly presenting the data image. We have added white arrows in the revised **Figure 5C** (original Figure 5B) to highlight that *BEERBELLY-OX* can influence the filamentous assembly of HeFP. According to the comment, we have rephrased the description as “To investigate the effect of *BEERBELLY* on the interaction between HeFP and tubulin, we co-expressed *BEERBELLY*, the filamentous distribution of HeFP in the cytoplasm disappeared (revised **Fig. 5C**, down panel), implying that *BEERBELLY* negatively regulated the binding of HeFP to tubulin.”(page 12, lines 360-363).

We have also rephrased the figure legend of revised **Figure 5D** (original Figure 5C) to make it more clearly: “The result showed that HA-tagged tubulin immunoprecipitated more Myc-tagged tubulin when HeFP was overexpressed (indicated by a red arrow), suggesting that HeFP facilitates microtubule polymerization” (**Figure legend of revised Figure 5D, page 27, lines 787-792**). Thank you.

Comment 7. Fig. 6D is capital to support the main conclusion on competition of HeFP. How many replicates of this experiment? Can this competition be observed in vivo on the Oex lines (preventing the binding of HeFP) to tubulin?

Reply: Three replicates of the experiments were performed in the study. As suggested, we have also used *HeFP-OX* plant to perform additional experiment to further confirm the conclusion. As the lack of an effective and specific antibody against HeFP, we cannot detect endogenous HeFP proteins. Instead, we used *HeFP-OX* protoplast cells which spontaneously express HeFP with a GFP tag, and detected the interaction of HeFP with the endogenous tubulin using anti-tubulin when transfected *BEERBELLY* or not. The result showed that *BEERBELLY* inhibits the interaction between endogenous HeFP and tubulin (see **below Fig. 16**). We have added the data to the revised manuscript (**page 12, lines 356-360**, marked in blue) and the revised **Figure 6D** (down panel) accordingly. Thanks for the comment.

Figure 16. Competitive binding assay of *BEERBELLY* in *HeFP-OX* protoplast cells.

Comment 8. Authors suggested that BEERBELLY is transcriptionally regulated by H3K27me3 during endosperm development. This is supported only by correlative analysis of ChIP-QPCR. To be more convincing I suggest the authors (i) to use public data bases to analyze the level of H3K27me3 in WT and in different polycomb mutants on this promoter and (ii) if one mutant displays a difference on BEERBELLY locus to do an RT-QPCR to check its transcription level.

Reply: We agree and have used public data from WT and a polycomb mutant *emf2a* to analyze the level of H3K27me3 on the promoter regions of *BEERBELLY*^{A,5}. As shown in the revised **Figure S7C** (also see **below Fig. 17**), the H3K27me3 modification was detected on -2.5kb upstream of *BEERBELLY* and down-regulated in the *emf2a* mutant. In addition, we obtained an *emf2a* plant and found that the transcription level of *BEERBELLY* was up-regulated in *emf2a* mutant. The data further supported our observation. We have added the results in the revised manuscript (**page 13, lines 385-389; pages 13-14, lines 396-398**, marked in blue) and **Figure S7C-D**. Thank you again for all your suggestions.

Figure 17. The H3K27me3 modification and *BEERBELLY* expression in *emf2a* mutant. **A.** The level of H3K27me3 modification on -2.5kb upstream of *BEERBELLY* in WT and *emf2a* endosperm; **B.** Relative expression level of *BEERBELLY* in WT and *emf2a* ($n = 3$ replicates, normalized against *ACTIN2*).

References cited in the point-by-point response

1. Lafon-Placette, C. & Kohler, C. Embryo and endosperm, partners in seed development. *Curr Opin Plant Biol.* **17**, 64-9 (2014).
2. Otegui, M.S., Mastrorarde, D.N., Kang, B.H., Bednarek, S.Y. & Staehelin, L.A. Three-dimensional analysis of syncytial-type cell plates during endosperm cellularization visualized by high resolution electron tomography. *Plant Cell.* **13**, 2033-51 (2001).
3. Wu, C. et al. Development of enhancer trap lines for functional analysis of the rice genome. *Plant J.* **35**, 418-27 (2003).
4. Tonosaki, K. et al. Mutation of the imprinted gene OsEMF2a induces autonomous endosperm development and delayed cellularization in rice. *Plant Cell.* **33**, 85-103 (2021).
5. Cheng, X. et al. The maternally expressed polycomb group gene OsEMF2a is essential for endosperm cellularization and imprinting in rice. *Plant Commun.* **2**, 100092 (2021).

REVIEWERS' COMMENTS

Reviewer #1 (Remarks to the Author):

The manuscript is the revised version of previously submitted work. All the questions from reviewers have been properly addressed and the manuscript has been extensively improved. I think it can be accepted as it is.

Reviewer #2 (Remarks to the Author):

The authors have addressed all my comments in an elegant way and performed several additional experiments to deal with each comment. I think the paper is significantly improved and it is a very nice story.